# When Tabular Foundation Models Meet Strategic Tabular Data: A Prior Alignment Approach

Xinpeng Lv [* 1]  Yunxin Mao [* 1]  Renzhe Xu [2]  Chunyuan Zheng [3]  Yikai Chen [1]  Haoxuan Li [3]  Jinxuan Yang [4]
Yuanlong Chen [5]  Kun Kuang [6]  Mingyang Geng [1]  Shixuan Liu [1]  Wanrong Huang [1]  Shaowu Yang [1]
Wenjing Yang [1]  Zhouchen Lin [3]  Haotian Wang [1]

## Abstract

Tabular foundation models based on pretrained prior-data fitted networks (PFNs) have shown strong generalization on diverse tabular tasks, but they are typically designed for *non-strategic* settings where data distributions are independent of deployed classifiers. In many real-world decision scenarios, however, individuals may strategically modify their features after deployment to obtain favorable outcomes, inducing a post-deployment distribution shift. This paper studies whether PFN-style tabular foundation models can generalize to such *strategic* tabular data. We show that strategic manipulation creates a fundamental mismatch between the non-strategic prior learned during pretraining and the post-manipulation strategic prior encountered at deployment, which leads to an irreducible structural prediction bias. To address this issue, we propose the **Strategic Prior-data Fitted Network** *(SPN)*, an inference-time strategy-aware framework that adapts tabular foundation models to strategic environments without retraining or architectural modification. SPN constructs strategic in-context examples to approximate post-manipulation inputs and aligns PFN predictions with the induced strategic distribution via in-context learning, with theoretical guarantees on bias reduction. Experiments on real-world and synthetic tabular datasets show that SPN consis-

tently improves robustness and predictive performance under strategic manipulation compared with both tabular foundation models and classical tabular methods.

## 1. Introduction

Tabular data is widely used across a broad range of real-world domains, including credit scoring, healthcare triage, and policy allocation (Borisov et al., 2022; Jagtiani & Lemieux, 2019; Sánchez-Monedero et al., 2020). Unlike vision or language domains, learning from tabular data requires reasoning over heterogeneous feature types, mixed categorical and numerical attributes, complex missingness patterns, and often severe class imbalance (Khosravi et al., 2023). Owing to these challenging data characteristics, tree-based methods remain the dominant class of models for tabular prediction.

Recently, this field has begun to shift to new possibilities to build tabular foundation models with the emergence of Prior-data fitted networks (PFNs) (Müller et al., 2024), such as TabPFN (Hollmann et al., 2025), TabICL (Qu et al., 2025), and TabDPT (Ma et al., 2025). By pre-training on a diverse distribution of tabular tasks, such models enable inference-time adaptation to new datasets via in-context learning (Jiang et al., 2026a). This amortized learning paradigm yields strong out-of-sample performance without task-specific retraining, marking a promising step toward general-purpose tabular learners.

Despite these advances, existing PFN-style tabular foundation models are almost exclusively developed and evaluated in *non-strategic* settings, where feature distributions remain fixed after deployment and data are observed passively (Gigerenzer, 2015). In many real-world decision pipelines, however, deployment occurs in *strategic* environments: once decision rules are known or inferred, individuals may actively adapt their observable features to obtain more favorable outcomes (Hardt et al., 2016; 2022). For example, as shown in Figure 1 for credit scoring, pretrained

[1]College of Computer, National University of Defense Technology, Changsha, China [2]Institute for Theoretical Computer Science, Shanghai University of Finance and Economics, Shanghai, China [3]State Key Lab of General AI, School of Intelligence Science and Technology, Peking University, Beijing, China [4]Faculty of Engineering, University of Sydney, Sydney, Australia [5]Faculty of Computing, Harbin Institute of Technology, Harbin, China [6]School of Artificial Intelligence, Zhejiang University, Hangzhou, China. Correspondence to: Haotian Wang <wanghaotian13@nudt.edu.cn>.

*Proceedings of the 43rd International Conference on Machine Learning*, Seoul, South Korea. PMLR 306, 2026. Copyright 2026 by the author(s).

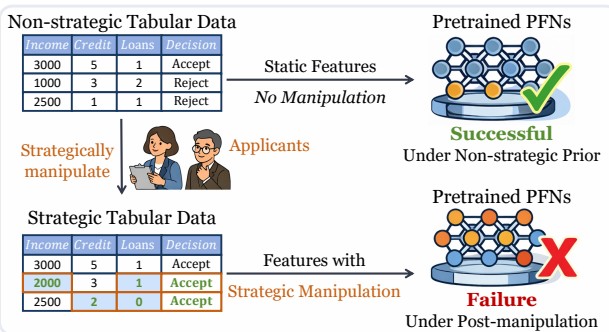

*Figure 1.* Illustration of strategic manipulation in tabular decision-making (e.g., credit scoring). PFNs perform well on non-strategic data but fail after deployment under strategic manipulation.

PFNs perform reliably when applicant features remain static. However, applicants may strategically adjust reported income or expenses to obtain favorable results (Milli et al., 2019), leading to systematic performance degradation.

Unfortunately, existing PFN-style tabular foundation models are not pretrained with such a strategic structure in mind, without accounting for agents' manipulations. Thus, this paper aims to close this gap by investigating the important but underexplored boundary of this family of models in the strategic regime:

> *Are PFN-style tabular foundation models capable of generalizing on strategic tabular data?*

Intuitively, PFN-style tabular models are pretrained under *non-strategic* settings that differ from the *strategic* settings at deployment, making systematic bias likely to arise. More concretely, in strategic settings, agents change their features in response to the model deployed, so the data observed at deployment no longer follows the same pattern as the training data. This means that the tabular model is asked to make predictions on feature configurations that arise *because of* strategic manipulation, rather than from the original data distribution it was trained on. Therefore, we show that the non-strategic prior learned during pretraining is misaligned with the post-manipulation distributions that arise in strategic environments, leading to a systematic prediction bias (see Section 4).

To address this challenge, our perspective is that the core challenge is not model capacity, but the *structure of inference*. Instantiating this idea with PFNs, we propose **Strategic Prior-data Fitted Networks** *(SPN)*, a two-stage inference framework that (inner stage) simulates agents' strategic manipulation through in-context interactions and (outer stage) aligns predictions with the induced post-manipulation distribution. Our SPN extends tabular foundation models from non-strategic prediction to a strategic (game-theoretic

decision) setting *without retraining or architectural modification*. **Our primary contributions are as follows:**

- We **characterize the boundary of PFN-style tabular priors in strategic environments** by identifying a fundamental mismatch between their *non-strategic* pretraining and the *strategic tabular data* encountered at deployment, and we prove that this misalignment induces a structural prediction bias.

- We introduce *Strategic Prior-Data Fitted Networks* (SPN), a strategy-aware, inference-time framework that aligns pretrained PFN-style models with strategic tabular data by leveraging the in-context learning capability of PFNs.

- We evaluate SPN on real-world and synthetic tabular data and show that it consistently improves performance under strategic settings, while retaining strong accuracy in non-strategic settings.

## 2. Related work

### 2.1. Tabular Data Learning

**Classical tabular models.** Tabular data learning is central to healthcare, finance, and social sciences, where tree ensembles such as XGBoost, LightGBM, and Cat-Boost have long dominated (Chen & Guestrin, 2016; Prokhorenkova et al., 2018). Neural approaches introduce tabular-specific inductive biases, including attentive feature selection (TabNet) (Arik & Pfister, 2021), differentiable trees (NODE) (Popov et al., 2019), and tree neural hybrids (Hazimeh et al., 2020). More recent work improves tabular representations via self-supervised learning (VIME, SCARF) (Yoon et al., 2020; Bahri et al., 2021) and Transformer-based architectures (TabTransformer, SAINT, NPT) (Huang et al., 2020; Somepalli et al., 2021; Kossen et al., 2021). These models, however, are typically trained per dataset and require task-specific optimization.

**Tabular foundation models.** Prior-data fitted networks (PFNs) (Hollmann et al., 2022) represent a major step toward *tabular foundation models*. TabPFN (Hollmann et al., 2022; 2025) is pretrained on large-scale synthetic tasks (e.g., SCM-simulated data) and performs prediction via in-context learning over the observed training table (Ma et al., 2025; Helli et al., 2024). Recent extensions improve data realism (Garg et al., 2025), robustness via lightweight adaptation/ensembling (Liu et al., 2025a), and scalability of tabular in-context learning (Qu et al., 2025). Related threads include a comprehensive survey of tabular representation learning (Jiang et al., 2026a) and LLM-based tabular reasoning for instance-wise ensembling and multimodal table understanding (Liu et al., 2025b; Jiang et al., 2026b). More related work is deferred to Appendix A.

## 2.2. Learning with Strategic Tabular Data

Deployed classifiers in tabular decision pipelines often induce strategic manipulation from individuals, giving rise to *strategic classification* (Hardt et al., 2016). This problem has been widely studied in settings with interpretable and partially manipulable features. Prior work largely focuses on learning classifiers that are robust to strategic behavior, typically assuming known or partially known manipulation models and relying on task-specific training or iterative optimization (Dong et al., 2017; Shavit et al., 2020; Chen et al., 2020; Harris et al., 2021; Zrnic et al., 2021; Wang et al., 2022; 2023; Tsirtsis et al., 2024; Shao et al., 2024; Ghalme et al., 2021). More recent approaches incorporate causal structure to distinguish genuine improvement from superficial manipulation, refining how strategic responses are modeled in tabular domains (Miller et al., 2020; Chen, 2023; Horowitz & Rosenfeld, 2023; Vo et al., 2024; Chang et al., 2024; Lv et al., 2026c; Wang et al.; 2025; Liu et al., 2026; Lv et al., 2026b).

## 3. Preliminary

Throughout this paper, we denote random variables by uppercase letters (e.g., $X$ and $Y$) and their realizations by lowercase letters (e.g., $x$ and $y$). Bold symbols (e.g., $\mathbf{x}$ and $\mathbf{X}$) are used for vectors or matrices.

### 3.1. Tabular Foundation Models: In-context Tabular Learning with PFNs

TabPFN (Hollmann et al., 2022) exemplifies tabular foundation models that solve supervised tabular tasks via *in-context* learning: conditioning on a labeled context table $\mathcal{D} = \{(x_i, y_i)\}_{i=1}^m$, the model predicts the label of a query point $x$ without updating parameters.

Formally, PFNs are pretrained across tasks sampled from a *non-strategic* meta-distribution $\hat{\Pi}$ over data-generating distributions $P$ on $(X, Y)$. For each task, $\mathcal{D} \sim P^m$ and $(x, y) \sim P$ are sampled i.i.d. from the same $P$. A PFN implements an inference map

$$\Phi_\theta : (\mathcal{D}, x) \mapsto \hat{y}, \qquad (1)$$

and is trained to minimize the expected prediction risk

$$J(\theta) = \mathbb{E}_{P\sim\hat{\Pi}}\mathbb{E}_{\mathcal{D}\sim P^m}\mathbb{E}_{(x,y)\sim P}[\mathcal{L}(\Phi_\theta(\mathcal{D}, x), y)]. \quad (2)$$

### 3.2. Strategic Tabular Data

A key difference between *non-strategic* and *strategic* tabular data is that the deployed decision rule *effects the input distribution*: individuals adjust their features in response to the classifier, creating a model-dependent shift (Hardt et al., 2016; Shao et al., 2024).

**Strategic manipulation.** When the decision maker deploys a scoring rule $f : \mathbb{R}^d \to \mathbb{R}$, agents with original features $x$ chooses a modified representation by strategic manipulation:

$$b_f(x) \in \arg\max_{x'\in\mathbb{R}^d} \left[f(x') - \lambda\, c(x, x')\right], \qquad (3)$$

where $c(x, x')$ is a manipulation cost and $\lambda > 0$ controls the cost–benefit trade-off.

As a result, evaluation is performed on the induced post-manipulation distribution, and the decision maker seeks a rule that is robust to strategic behavior:

$$f^* \in \arg\min_{f\in\mathcal{F}}\mathbb{E}_{(x,y)}[\mathcal{L}(f(b_f(x)), y)]. \qquad (4)$$

### 3.3. In-Context Learning as Implicit Gradient Descent

PFNs produce predictions by conditioning on a labeled context table $\mathcal{D}$ and a query $x$ in a single forward pass. Although model parameters remain fixed, self-attention repeatedly integrates information from the context into the query representation across layers, yielding an evolving internal state. This layer-wise evolution can be interpreted as a form of (preconditioned) gradient descent on an implicit objective induced by the context (Akyürek et al., 2022; Ahn et al., 2023; Von Oswald et al., 2023).

**Lemma 3.1** (Forward pass as implicit optimization (Akyürek et al., 2022; Ahn et al., 2023))**.** *Consider a Transformer conditioned on a context $\mathcal{D} = \{(x_i, y_i)\}_{i=1}^n$ and a query $x$. There exists a sequence of internal states $\{w_\ell\}_{\ell=1}^L$ such that the layer-$\ell$ output admits a linear readout*

$$\hat{y}_\ell(x) = -\langle x, w_\ell \rangle, \qquad (5)$$

*and the state evolves across layers as*

$$w_{\ell+1} = w_\ell - A_\ell \nabla R_\mathcal{D}(w_\ell), \qquad (6)$$

*where $R_\mathcal{D}$ is an implicit objective determined by the context (e.g., a least-squares risk over $\{(x_i, y_i)\}_{i=1}^n$), and $A_\ell$ is a layer-dependent preconditioning (step) matrix.*

Intuitively, each Transformer layer performs an optimization step, allowing the forward pass to emulate gradient-based adaptation without parameter updates (see Appendix B).

## 4. Theory: The Risk of Failure by PFNs on Strategic Tabular Data

In this section, we distinguish learning on tabular data under two regimes: the *non-strategic setting* (Shwartz-Ziv & Armon, 2022; Ucar et al., 2021) and the *strategic setting* (Hardt et al., 2016; 2022). We then analyze how a PFN pretrained under a non-strategic meta-prior can suffer *structural* failure when it is directly adapted to strategic tabular tasks.

## 4.1. Non-strategic vs Strategic Learning on Tabular Data

**Non-strategic setting.** In non-strategic settings, individuals do not modify their features in response to a model or decision rule.

**Definition 4.1** (Non-strategic Setting). *A non-strategic setting* $\Pi_{\text{non-strategic}}$ *is a meta-distribution over tabular task distributions in which agents do not strategically manipulate their features. For each task, a distribution $P$ over $(X, Y)$ is drawn as $P \sim \Pi_{\text{non-strategic}}$, and samples are generated as $(X, Y) \sim P$.*

**Strategic setting.** In strategic settings, individuals strategically modify their features to obtain favorable outcomes, so the data distribution changes after a classifier is deployed.

**Definition 4.2** (Strategic Setting). *A strategic setting* $\Pi_{\text{strategic}}$ *is a meta-distribution in which agents' strategic modifications of features alter the data distribution after deployment. For a given task, after a classifier $f$ is deployed, agents update their features according to a response map $b_f$, yielding observed samples*

$$(X', Y) = (b_f(X), Y). \tag{7}$$

## 4.2. Risk of Directly Adapting PFNs on Strategic Tabular Data: Meta-prior Mismatch

We formalize how a mismatch between the non-strategic meta-prior learned during PFN pretraining and the true meta-prior in strategic settings can lead to prediction bias.

**Pretraining Occurs Under a Non-strategic Meta-Prior.** During pretraining, each task $P$ is sampled from a meta-prior $\hat{\Pi}$ in which all data distribution are from non-strategic settings $\Pi_{\text{non-strategic}}$:

$$P \sim \hat{\Pi}, \qquad \hat{\Pi} \subset \Pi_{\text{non-strategic}}. \tag{8}$$

**Deployment Operates Under a Strategic Meta-Prior.** However, in the strategic regime, agents modify their features in response to the deployed classifier $f$. The PFN is therefore evaluated on the corresponding *post-manipulation* task distributions:

$$P_f^{\text{strategic}} \sim \Pi_{\text{strategic}}, \;\; \Pi_{\text{strategic}} \not\subseteq \Pi_{\text{non-strategic}}. \tag{9}$$

In other words, training observes the original distributions $P$, whereas deployment must operate on strategically transformed distributions $P_f^{\text{strategic}}$, creating a potential *meta-prior mismatch* between the pretraining environment and the strategic environments encountered at test time.

**Uncovered strategic distributions and TV mismatch.** For a meta-prior $\Pi$ over task distributions, we denote $\text{supp}(\Pi)$

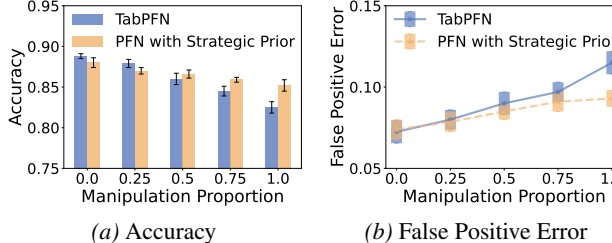

*(a)* Accuracy      *(b)* False Positive Error

*Figure 2.* Performance of TabPFN and SPN under increasing strategic manipulation. (a) Accuracy and (b) false positive error as the proportion of manipulated inputs increases.

as the set of task distributions that occur with non-zero probability under $\Pi$. Among the strategic task distributions $P \in \text{supp}(\Pi_{\text{strategic}})$, some may lie entirely outside the support of the non-strategic meta-prior $\Pi_{\text{non-strategic}}$. We formalize such *uncovered strategic distributions* in

$$\mathcal{S}_0^{\text{stra}} := \left\{ P \in \text{supp}(\Pi_{\text{strategic}}) : P \notin \text{supp}(\Pi_{\text{non-strategic}}) \right\}, \tag{10}$$

where $\text{supp}(\cdot)$ denotes the support of a distribution over task distributions.

**Quantifying the mismatch between meta-priors.** To characterize how severe this support mismatch is, we quantify the proportion of such out-of-support tasks under the strategic prior, i.e., the corresponding *uncovered strategic mass*:

$$\delta := \Pi_{\text{strategic}}(\mathcal{S}_0^{\text{stra}}), \tag{11}$$

which measures the probability that a task sampled from $\Pi_{\text{strategic}}$ lies outside the support of $\Pi_{\text{non-strategic}}$.

We further relate this uncovered mass to the total variation (TV) distance (Bhattacharyya et al., 2022) between the two priors, obtaining the following bound (see detailed proof in Appendix C.2).

**Lemma 4.3.** *The discrepancy between the strategic meta-prior and the non-strategic meta-prior satisfies*

$$\begin{aligned} \text{TV}\,(\Pi_{\text{strategic}}, \Pi_{\text{non-strategic}}) \geq \\ \left| \Pi_{\text{strategic}}(\mathcal{S}_0^{\text{stra}}) - \Pi_{\text{non-strategic}}(\mathcal{S}_0^{\text{stra}}) \right| = \delta, \end{aligned} \tag{12}$$

*where the inequality follows from the definition $\text{TV}(\mu, \nu) = \sup_A |\mu(A) - \nu(A)|$ by taking $A = \mathcal{S}_0^{\text{stra}}$, and the last equality uses $\mathcal{S}_0^{\text{stra}} \cap \text{supp}(\Pi_{\text{non-strategic}}) = \emptyset$, which implies $\Pi_{\text{non-strategic}}(\mathcal{S}_0^{\text{stra}}) = 0$.*

## 4.3. From meta-prior mismatch to prediction bias

Let $\psi(P)$ be a scalar prediction-relevant functional of a task distribution $P$, and let $\hat{\psi}_n$ be an estimator based on $n$ i.i.d. samples $Z_{1:n} \sim P$. For a random tabular task in strategic settings $P \sim \Pi_{\text{strategic}}$, we consider the average approximation error:

$$\mathcal{E}_n := \mathbb{E}_{P \sim \Pi_{\text{strategic}}} \mathbb{E}_{Z_{1:n} \sim P} \left[ |\hat{\psi}_n - \psi(P)| \right]. \tag{13}$$

In words, $\mathcal{E}_n$ describes how closely a procedure calibrated to the non-strategic meta-prior can approximate the quantity $\psi(P)$ across strategic tasks when given $n$ samples per task.

**Prediction bias.** From Eq. (11), let $\delta = \Pi_{\text{strategic}}(\mathcal{S}_0^{\text{stra}})$. Under a mild regularity condition on $\psi$ (see Proposition 4.4), there exists a constant $c > 0$ such that any sequence of estimators $\hat{\psi}_n$ satisfies

$$\liminf_{n \to \infty} \mathcal{E}_n \ \geq \ c\,\delta, \tag{14}$$

showing that any non-zero uncovered mass $\delta$ necessarily induces an *irreducible strategic bias* (see detailed proof of in Appendix C.3).

**Proposition 4.4** (Unavoidable Strategic Bias). *Let $\mathcal{S}_0^{\text{stra}}$ and $\delta$ be defined as in Eq. (11), and assume $\Pi_{\text{non-strategic}}(\mathcal{S}_0^{\text{stra}}) = 0$. Suppose that $\psi$ is Lipschitz with respect to the total variation distance and that there exists a margin $\gamma > 0$ such that*

$$\begin{aligned} |\psi(P) - \psi(Q)| \ &\geq \ \gamma \\ \text{for all} \quad P \in \mathcal{S}_0^{\text{stra}}, \ &Q \in \text{supp}(\Pi_{\text{non-strategic}}). \end{aligned} \tag{15}$$

*Then there exists a constant $c > 0$, such that for any sequence of estimators $\hat{\psi}_n$,*

$$\liminf_{n \to \infty} \mathcal{E}_n \ \geq \ c\,\delta. \tag{16}$$

As shown in Figure 2, the PFNs under the non-strategic prior decrease in accuracy, while false positive errors increase with the increasing proportion of strategic manipulation.

# 5. Our Method: Strategic Prior-data Fitted Networks

In strategic settings, individuals modify their features in response to the deployed classifier $f$. Let $b_f$ denote the best-response function and $(X', Y) = (b_f(X), Y)$ with induced distribution $P_f^{\text{strategic}}$. We thus define the *strategic risk* as

$$R_{\text{strategic}}(f; P) = \mathbb{E}_{(X',Y) \sim P_f^{\text{strategic}}}\big[\mathcal{L}(f(X'), Y)\big]. \tag{17}$$

To address this risk, we consider two principled methods: (i) *parameter updating via fine-tuning*, which explicitly updates the model to match strategic tabular data; (ii) *inference-time alignment* via in-context learning (ICL).

In Section 5.1, we consider fine-tuning as a direct baseline for mitigating strategic risk. Due to rapidly evolving strategic manipulations (Mendler-Dünner et al., 2020; Lv et al., 2026a), repeated finetuning leads to significant computational overhead. Motivated by this limitation, in Sections 5.2 and 5.3 we explore the ability of in-context learning (ICL) to support inference-time alignment, proposing *Strategic Prior Alignment*.

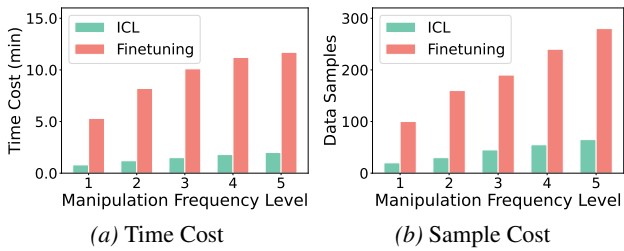

*(a)* Time Cost       *(b)* Sample Cost

*Figure 3.* A case study comparing the time and data costs of ICL and finetuning across increasing levels of manipulation frequency. Levels indicate increasing manipulation frequency, from sparse to dense regimes.

## 5.1. A Case Study: Practical Cost of Finetuning vs. ICL

We first consider fine-tuning as a direct baseline for reducing the strategic risk $R_{\text{strategic}}(f; P)$. Specifically, given samples $\{(x_i, y_i)\}_{i=1}^n$ and a manipulation function $b_f$ induced by the deployed classifier, we construct the augmented strategic tabular data $\mathcal{D}_{ft} := \{(x_i, y_i)\}_{i=1}^n \cup \{(b_f(x_i), y_i)\}_{i=1}^n$.

**A semi-synthetic case study grounded in real-world data.** We conduct a semi-synthetic case study based on real-world email spam (Heydari et al., 2015) to compare the practical cost between deploying finetuning and using ICL in strategic settings. Prior work (Jáñez-Martino et al., 2023; Henke et al., 2021) has shown that spam filtering involves frequent strategic manipulation, requiring classifiers to be updated repeatedly. Accordingly, we use real-world spam data with simulated strategic manipulations (Zrnic et al., 2021; Chen, 2023), varying the manipulation frequency as real-world attackers.

Following prior work on fine-tuning and ICL (Mosbach et al., 2023; Yin et al., 2024), we evaluate both methods under increasing manipulation frequency using two operational cost metrics (see Appendix D for details):

- **Update time cost**: wall-clock time required per update cycle, capturing the parameter-update overhead;

- **Update data cost**: the number of strategic (manipulated) samples consumed per update cycle.

As shown in Fig. 3, the cost of fine-tuning increases rapidly as manipulation becomes more frequent, since each update requires additional training and newly collected strategic data. In contrast, ICL adapts without parameter updates, leading to substantially lower time and data costs across frequencies.

Since strategic manipulation typically requires repeated updates rather than a one-time correction, the overhead of fine-tuning accumulates quickly in practice. Building on this insight, we develop a method that leverages in-context learning to reduce strategic risk without additional training.

## 5.2. In-context Strategic Manipulation

We now describe how to align a pretrained PFN with strategic tabular data at inference time. A PFN produces predictions conditional on both the query $x$ and the context $\mathcal{D}$: e.g., $f_\theta^{(PFN)}(x \mid \mathcal{D})$. Because predictions are inferred from attention-based interactions over the context, modifying the context also effects the predictions of PFNs.

**Strategic tabular context construction.** Rather than finetuning PFNs, we construct a *strategic tabular context* by pairing each observed feature vector with its post-manipulation counterpart. This paired context enables attention-based in-context learning to implicitly adapt predictions to the strategic setting at inference time

**Definition 5.1** (Strategic Tabular Context)**.** Given an observed tabular example $z_i := (x_i, y_i)$ from the non-strategic environment, we simulate the post-manipulation counterpart $z_i'$

$$z_i' := (b_f(x_i), y_i), \tag{18}$$

where $b_f(\cdot)$ denotes the agent best-response mapping induced by the deployed rule $f$. Each pair $\{z_i, z_i'\}$ represents a single strategic context example, and a collection of such pairs constitutes a **strategic tabular context**:

$$\widetilde{\mathcal{D}} = \{z_i, z_i'\}_{i=1}^n. \tag{19}$$

**Strategic alignment via attention.** Given the strategic context $\widetilde{\mathcal{D}}$, a pretrained PFN performs in-context learning via attention. As stated in Lemma 3.1, this attention updating can be interpreted as an implicit, gradient-like strategic update. Therefore, ICL with $\widetilde{\mathcal{D}}$ aligns PFN inference with the strategic setting. In particular, we have the approximate alignment (see detailed proof in Appendix E):

**Proposition 5.2** (Strategic alignment via attention)**.** *Under the strategic tabular context $\widetilde{\mathcal{D}}$, the ICL-induced attention update of a pretrained PFN is aligned with the one-step gradient-based strategic manipulation update:*

$$\Delta_{\mathrm{ICL}}(x; \widetilde{\mathcal{D}}, f_\theta^{(\mathrm{PFN})}) \approx \Delta_{\mathrm{GD}}(x; f), \tag{20}$$

*where $\Delta_{\mathrm{ICL}}$ denotes the ICL-induced update and $\Delta_{\mathrm{GD}}$ denotes gradient-based strategic manipulation update.*

## 5.3. Inference-time Bi-level Optimization

Given the ICL-adjusted context $\widetilde{\mathcal{D}}_f$ constructed in Section 5.2, SPN performs prediction by running the PFN on this adjusted table. For a query $x^*$, the SPN predictor is

$$f_{\mathrm{SPN}}(x^*) = \Phi_{\mathrm{PFN}}^{(\mathrm{out})}(x^* \mid \widetilde{\mathcal{D}}_f), \tag{21}$$

where $\Phi_{\mathrm{PFN}}^{(\mathrm{out})}$ denotes the standard PFN forward pass. Since $\widetilde{\mathcal{D}}_f$ approximates how features would be manipulated in response to the deployed rule $f$, evaluating $f_{\mathrm{SPN}}$ on $(x^*, \widetilde{\mathcal{D}}_f)$

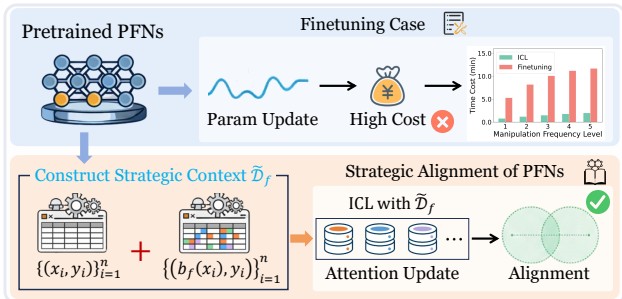

*Figure 4.* Overview of the SPN framework. SPN aligns PFN-style models to strategic environments at inference time.

effectively aligns the PFN's predictions with the strategic risk defined in Eq. (17).

This procedure simulates, at inference time, the bi-level structure for the strategic setting:

- **Inner stage — adapting strategic settings:** in-context inputs are modified according to anticipated agent responses, yielding the adjusted context $\widetilde{\mathcal{D}}_f$;

- **Outer stage — prediction under strategic settings:** the PFN predicts for a query $x^*$ by conditioning on $\widetilde{\mathcal{D}}_f$, i.e., $\hat{y} = \Phi_{\mathrm{PFN}}^{(\mathrm{out})}(x \mid \widetilde{\mathcal{D}}_f)$.

SPN mitigates this effect by adjusting the inference-time context to reflect strategic manipulations by agents, thereby reducing the prediction bias of PFNs in strategic settings.

**Proposition 5.3** (SPN reduces prediction bias (see detailed proof in Appendix G))**.** *Let $\Pi_{\mathrm{strategic}}^{\mathrm{SPN}}$ denote the inference-time task distribution induced by SPN. The corresponding uncovered mass under SPN is strictly smaller than the uncovered mass $\delta$ (in Proposition 4.4), i.e.,*

$$\delta_{\mathrm{SPN}} := \Pi_{\mathrm{strategic}}^{\mathrm{SPN}}(P \notin \mathrm{supp}(\Pi_{\mathrm{non\text{-}strategic}})) < \delta. \tag{22}$$

The whole process of the strategic prior-data fitted network is illustrated in Algorithm 1.

## 6. Experiment

We evaluate the proposed *Strategic Prior-data Fitted Network (SPN)* as an *inference-time* extension for PFNs in strategic classification. Specifically, our experiments address the following questions:

- How strategic manipulation affects non-strategic PFNs, and whether SPN can mitigate this degradation.

- Whether SPN preserves predictive performance on non-strategic tabular benchmarks.

- How SPN performs across different ICL scales and manipulation regimes.

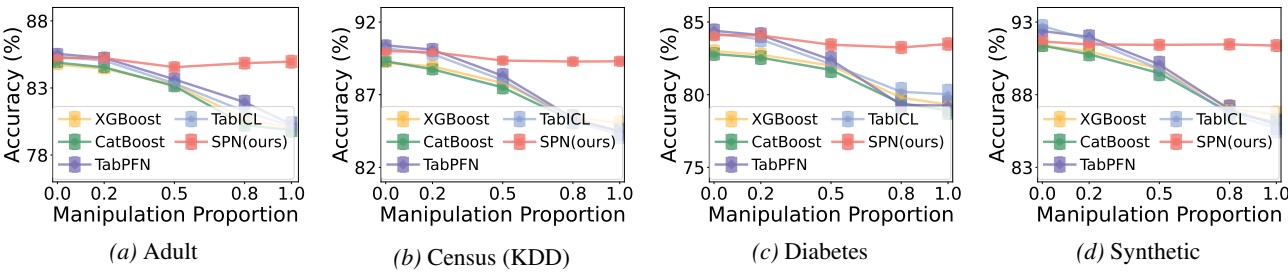

*Figure 5.* Performance of tabular models with different manipulation proportions across real-world and synthetic datasets.

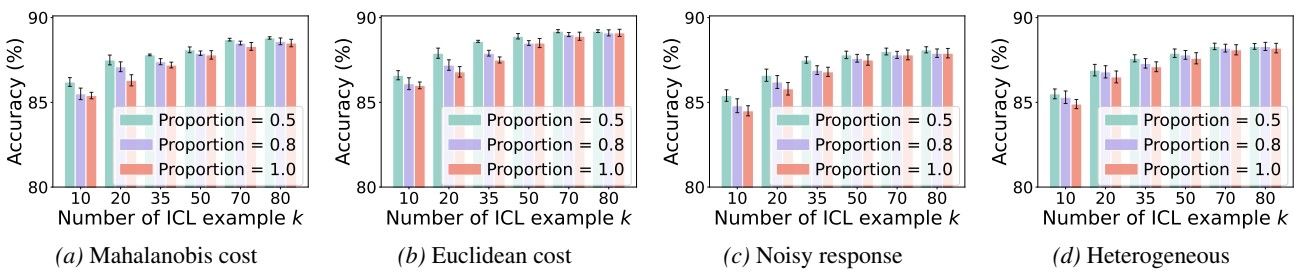

*Figure 6.* Effect of the ICL scale (number of in-context examples) under four different manipulation regimes (as shown in Section 6.2), evaluated at different manipulation proportions (0.5, 0.8, 1.0).

---

**Algorithm 1** Strategic Prior-data Fitted Network (SPN)

---

**Require:** A pretrained PFN $f_\theta^{(PFN)}$; original labeled data $\mathcal{D} = \{(x_i, y_i)\}_{i=1}^n$; strategic test set $\tilde{\mathcal{T}} = \{(\tilde{x}_j, y_j)\}_{j=1}^m$; manipulation function $b_f(\cdot)$; strategic context size $K$.

**Ensure:** Predictions $\{\hat{y}_j\}_{j=1}^m$ on $\tilde{\mathcal{T}}$.

1: **Strategic context construction:**
2: Select a subset $\mathcal{D}_k \subset \mathcal{D}$.
3: **for** each $(x_i, y_i) \in \mathcal{D}_k$ **do**
4:     Compute manipulated feature $\tilde{x}_i \leftarrow b_f(x_i; f_\theta^{(PFN)})$
5: **end for**
6: Form strategic tabular context $\tilde{\mathcal{D}} \leftarrow \{Pair((x_i, y_i), (\tilde{x}_i, y_i))\}_{i=1}^K$
7: PFN $f_\theta^{(PFN)}$ ICL with $\tilde{\mathcal{D}}$
8: **Inference on strategic test data:** $\hat{y}_j \leftarrow f_\theta^{(PFN)}(\tilde{\mathcal{T}})$
9: Return $\{\hat{y}_j\}_{j=1}^m$

---

## 6.1. Experimental Setup

**Datasets.** We evaluate our methods on two categories of tabular datasets, corresponding to distinct evaluation goals. First, we consider a set of *strategic benchmarks* commonly used in prior work on strategic tabular learning (Chen, 2023; Lv et al., 2026a), such as *Adult* and *Spambase*. Second, we report results on *non-strategic tabular benchmarks* (Jiang et al., 2026a; Majee et al., 2025) that do not involve feature manipulation, such as *Bank-Marketing* and *Phishing*. Detailed dataset descriptions are provided in Appendix H.1.

**Methods.** We compare the proposed SPN against two broad classes of tabular learning approaches. First, we consider *classical tabular models*, such as *XGBoost*, *LightGBM*, and so on. Second, we evaluate recent advanced *pretrained tabular foundation models* with in-context learning, such as *TabPFN*, *TabDPT*, and so on. Detailed models and descriptions are provided in Appendix H.2.

**Evaluation settings.** For in-context learning, the contexts of tabular models come from non-strategic data, and SPN constructs its contexts from strategic data. At inference time, under the *non-strategic setting*, all models perform inference on non-strategic inputs. Under the *strategic setting*, models are evaluated on strategic test distributions with 80% strategic and 20% non-strategic inputs.

**Implementation.** SPN is an inference-time framework on a pretrained TabPFN backbone. SPN constructs a strategic context $\tilde{\mathcal{D}}_f$ by pairing each observed labeled point $z_i$ with its post-manipulation counterpart $z_i'$. These context pairs are aggregated into a strategic tabular context $\tilde{\mathcal{D}}_f$, which is then used for in-context learning at inference time (see detailed examples in Appendix H.3).

## 6.2. Different Manipulation Regimes

To assess robustness beyond a single strategic model, we consider multiple manipulation regimes studied in the strategic classification literature. Each regime captures a distinct aspect of how individuals may respond to deployed decision rules. Formal definitions and details are included in Appendix H.4.

• **Mahalanobis-cost manipulation** (*Mah*). A canonical

*Table 1.* Performance of tabular foundation models and their strategic extensions on standard (non-strategic) tabular benchmarks. Results are reported as mean AUC-ROC (%).

| Model | Bank | Blood | Phishing | Heart | Car (binary) | Diabetes (US) | COIL 2000 | Tic-Tac-Toe |
|---|---|---|---|---|---|---|---|---|
| TabPFN v2.5 | 91.85 | 78.10 | 95.02 | 92.85 | 99.28 | 83.35 | 73.85 | 99.69 |
| Chunked TabPFN | 91.92 | 78.05 | 95.10 | 92.78 | 99.30 | 83.28 | 73.90 | 99.68 |
| Drift-Resilient TabPFN | 91.70 | 77.88 | 94.85 | 92.60 | 99.22 | 83.10 | 73.60 | 99.55 |
| TabDPT | 91.45 | 77.60 | 94.70 | 92.40 | 99.20 | 82.95 | 73.40 | 99.51 |
| TabICL | 91.52 | 77.72 | 94.82 | 92.48 | 99.18 | 83.02 | 73.55 | 99.58 |
| TabFlex | 91.88 | 78.02 | 95.08 | 92.75 | 99.26 | 83.30 | 73.92 | 99.62 |
| **SPN (ours)** | 91.65 | 77.82 | 94.80 | 92.57 | 99.22 | 83.15 | 73.55 | 99.66 |

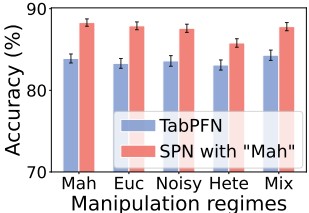 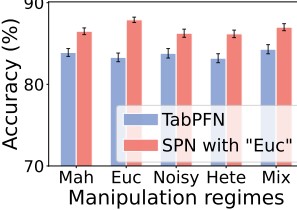 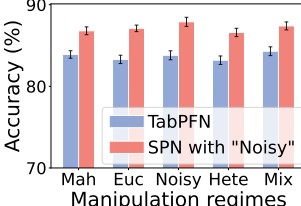 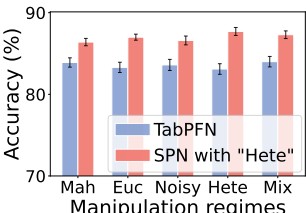

*(a)* SPN with "Mah" examples   *(b)* SPN with "Euc" examples   *(c)* SPN with "Noisy" examples   *(d)* SPN with "Hete" examples

*Figure 7.* Performance under different test-time manipulation regimes. *Mah*, *Euc*, *Noisy*, and *Hete* denote Standard manipulation with Mahalanobis cost, Standard manipulation with Euclidean cost, Noisy response, and Heterogeneous manipulation capability, respectively, while Mix denotes an equal mixture of all regimes and 20% non-manipulation. Each subfigure fixes the manipulation model used to construct ICL examples.

regime that models correlated feature manipulation via a Mahalanobis cost (Gavish et al., 2022; Chen, 2023).

- **Euclidean-cost manipulation** (*Euc*). A canonical regime assuming independent feature manipulation measured by Euclidean distance (Hardt et al., 2016; Zrnic et al., 2021).

- **Noisy strategic manipulation** (*Noisy*). Models imperfect feedback or bounded rationality through noisy evaluations of the deployed rule (Levanon & Rosenfeld, 2021; Ghalme et al., 2021).

- **Heterogeneous manipulation capability** (*Hete*). Captures population heterogeneity by allowing individual-specific manipulation costs (Shao et al., 2024).

### 6.3. Results and Analysis

**Prediction under strategic settings.** Prediction under strategic settings. Figure 5 reports accuracy as the manipulation proportion increases on both real-world and synthetic datasets. In this experiment, the manipulation proportion controls the overall strength of strategic influence at the dataset level, i.e., the fraction of test samples that are replaced by their strategically manipulated counterparts. Therefore, a larger manipulation proportion corresponds to a stronger post-deployment distribution shift induced by strategic behavior.

Across all four datasets, standard tabular models and tabular foundation models (e.g., TabPFN and TabDPT) exhibit clear performance degradation as the manipulation strength increases. This indicates that vanilla models are sensitive to stronger strategic shifts and tend to suffer from increasing prediction bias when more individuals manipulate their features. In contrast, SPN remains markedly more stable across different manipulation proportions andconsistently outperforms the competing baselines, especially in high-manipulation regimes. These results suggest that the strategic context constructed by SPN can effectively mitigate the impact of increasing manipulation strength.

**Effect of the number of in-context examples.** Figure 6 studies how the number of in-context examples $k$ affects SPN performance under strategic inference across four manipulation regimes (Section 6.2). Across all regimes and manipulation proportions, increasing $k$ consistently improves accuracy, with the largest gains occurring when $k$ increases from 10 to around 50. Beyond this range, performance improvements become marginal, indicating a clear saturation effect. Importantly, this trend is stable across different manipulation proportions (0.5, 0.8, and 1.0), suggesting that SPN's correction of strategic distribution shift relies on a moderate amount of strategic context rather than large in-context tables. Overall, these results indicate that SPN is sample-efficient in its use of in-context examples and robust

across diverse strategic response models.

**Performance under non-strategic settings.** Table 1 reports mean AUC-ROC on standard non-strategic tabular benchmarks. Despite being designed for strategic robustness, SPN does not sacrifice performance in the absence of manipulation: across all datasets, SPN achieves results that are comparable to strong pretrained tabular foundation models (e.g., TabPFN v2.5, TabDPT, and TabFlex), with only minor differences. Notably, this holds even though SPN still constructs its in-context inputs using a *strategic* tabular context at inference time, while evaluation queries are drawn from the original (non-strategic) test distribution. These results show that SPN preserves the generalization behavior of pretrained tabular foundation models under non-strategic evaluation.

**Generalization across manipulation regimes.** Figure 7 evaluates SPN under mismatched manipulation regimes by constructing strategic in-context examples using a single manipulation regime (Mahalanobis or Euclidean), while testing on a range of alternative regimes. Across all settings, SPN consistently outperforms TabPFN, including under noisy, heterogeneous, and mixed manipulation. These results show that, when guided by strategic in-context information, SPN maintains robust performance across different manipulation regimes, even when the inference-time strategic test inputs differ from those used to construct the context.

More experimental results are included in Appendix H.6

## 7. Conclusion

We study PFN-style tabular foundation models in strategic environments, where deployed decision rules induce endogenous, post-deployment distribution shifts through agents' feature manipulations. We show that this setting creates a fundamental mismatch between the non-strategic meta-prior learned in pretraining and the post-manipulation task distributions, leading to systematic prediction bias and performance degradation under manipulation. To bridge this gap, we propose strategic prior-data fitted networks, an inference-time prior-alignment framework that anticipates strategic responses by constructing strategic in-context examples, requiring neither retraining nor architectural changes.

## Acknowledgement

This work was jointly supported by the National Natural Science Foundation of China (Nos. 62276004, 62325604, 62525213), the Beijing Natural Science Foundation (No. L257007), the Beijing Major Science and Technology Project (No. Z251100008425006), the Shanghai Sailing Program (No. 24YF2711600), the Provincial Natural Science Foundation of Heilongjiang Province (No. LH2023C069), the Provincial Natural Science Foundation of Hunan Province (No. 2025JJ10008), the NUDT Youth Independent Innovation Science Fund (No. ZK25-20), the NUDT Innovation Foundation for Postgraduate (No. XJQY2025011), and the State Key Laboratory of General Artificial Intelligence.

## Impact Statement

This paper aims to advance machine learning for strategic tabular decision-making, where individuals may adapt their features in response to deployed classifiers. Such settings are common in socially relevant domains, such as credit scoring, resource allocation, and policy screening. By improving the robustness of tabular foundation models under strategic manipulation, our work may help reduce prediction errors caused by post-deployment distribution shifts.

At the same time, strategic decision systems may have important societal implications. If the manipulation model is misspecified, or if individuals differ in their access to information and ability to respond strategically, the resulting decisions may still lead to unfair outcomes across groups. Therefore, SPN should not be viewed as a fully automated solution for high-stakes decision-making. In practical deployments, it is important to evaluate model behavior under diverse strategic assumptions, account for heterogeneous response capabilities, and carefully monitor fairness and resource allocation outcomes before use.

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

## Appendix Contents

## A. Additional Related Work: Tabular Data Learning From Classical Models to Foundation Models

### A.1. Classical Tabular Models.

Tabular prediction is a core workload in domains such as healthcare and finance, where tree ensembles remain strong and widely adopted baselines. Gradient boosted decision trees (GBDTs), including XGBoost, LightGBM, and CatBoost, are consistently competitive across heterogeneous feature types and moderate data regimes (Chen & Guestrin, 2016; Prokhorenkova et al., 2018; Ke et al., 2017). Their robustness and ease of tuning have made them a fact reference point for tabular benchmarks and practical deployments.

### A.2. Deep Learning for Tabular Data: Inductive Biases and Architectures.

To bridge the performance gap between deep models and GBDTs, prior work introduced tabular-specific inductive biases: attentive feature selection and interpretability (TabNet) (Arik & Pfister, 2021), differentiable tree-style computation (NODE) (Popov et al., 2019), and hybrid tree–neural designs (Hazimeh et al., 2020). Transformer-style architectures later became prominent by treating features as tokens and modeling feature interactions explicitly, including TabTransformer (Huang et al., 2020), SAINT (Somepalli et al., 2021), and non-parametric or set/sequence style Transformers for tabular inputs (e.g., NPT) (Kossen et al., 2021). A complementary line of work re-examined baselines and training protocols, showing that simple MLP/ResNet variants and a feature-tokenizing Transformer (FT-Transformer) are strong and often under-appreciated baselines on common benchmarks (Gorishniy et al., 2021).

### A.3. Self-supervised Representation Learning and Transfer for Tabular Data.

Beyond supervised training per dataset, representation learning has been explored to improve label efficiency and transferability. Methods include predictive/self-supervised objectives tailored to tabular corruption or masking (e.g., VIME) (Yoon et al., 2020; Liu et al., 2024; Chen et al., 2026) and contrastive learning via random feature corruption (SCARF) (Bahri et al., 2021; Kou et al., 2025). These approaches generally pretrain a representation on unlabeled or weakly labeled data and then fine-tune for a specific downstream dataset/task, offering a different pathway toward generalization than "train-from-scratch" tabular DL.

### A.4. Tabular Foundation Models and Universal Tabular Learners.

A major recent shift is the emergence of *tabular foundation models*: models designed to generalize across datasets/tasks with minimal or no gradient-based adaptation. Prior-data fitted networks (PFNs) (Hollmann et al., 2022) operationalize this idea by pretraining a model on a distribution over *tasks/datasets* (often synthetically generated) so that *conditioning on a context table* at test time approximates Bayesian inference under the learned prior. TabPFN (Hollmann et al., 2022; 2025) is a representative PFN instantiation, pretrained on large-scale synthetic tasks (e.g., SCM-simulated datasets) and performing prediction via in-context learning (context data + query) without test-time gradient updates. Subsequent work continues

to extend this paradigm along multiple directions, such as improving realism by incorporating real-data pretraining (Garg et al., 2025), enhancing robustness through lightweight adaptation or ensembling (Liu et al., 2025a), and scaling tabular in-context learning to larger tables or longer contexts (Qu et al., 2025). At a higher level, recent surveys systematize tabular representation learning and discuss foundation models as "general" tabular learners that unify transfer, robustness, and evaluation protocols (Jiang et al., 2026a; Kou et al., 2026).

### A.5. Tabular LLMs with In-context Learning

In parallel to PFN-style models that are *native* to table inputs, another research thread leverages large language models by serializing tabular data into text prompts for zero-/few-shot prediction or reasoning (Gardner, 2024; Zhang et al., 2023). Recent work further explores *LLM-centered tabular pipelines* where the LLM is used as a reasoning/aggregation module (e.g., instance-wise ensembling driven by chain-of-thought style prompting) (Liu et al., 2025b), and more broadly evaluates progress and limitations via tabular leaderboards and prompting-based reasoning traces. In particular, Ye and collaborators provide a comprehensive survey of tabular representation learning, and propose/benchmark chain-of-thought style reasoning and ensembling on tabular leaderboards (Jiang et al., 2026a; Liu et al., 2025b). These LLM-based lines are typically complementary to PFNs: they emphasize language-mediated reasoning and tool/ensemble integration, whereas PFNs emphasize learning a task prior that enables fast amortized inference directly from context tables.

### A.6. Positioning of Our Work.

Most tabular foundation models (PFNs and their extensions) and LLM-based tabular reasoning methods are developed and evaluated under *non-strategic* assumptions, where data are treated as passive observations. In contrast, our focus is to understand how PFN-style priors and tabular in-context learning behave when the data become *strategic* and decision-dependent, and to benchmark strategic settings against both classical tabular models and modern tabular foundation models.

## B. Implicit Gradient Descent in Self-Attention Layers

Our Lemma 3.1 indicates that, under ICL guidance, the token update process within the self-attention layer can be viewed as an implicit gradient optimization process (Akyürek et al., 2022).

First, we highlight the dependency on the tokens $e_i$ of the linear self-attention operation

$$
\begin{aligned}
e_j \leftarrow e_j + \text{SA}(\{e_1, \dots, e_N\}) &= e_j + \sum_h P_h V_h K_h^T q_{h,j} \\
&= e_j + \sum_h P_h \sum_i v_{h,i} \otimes k_{h,i} q_{h,j} \\
&= e_j + \sum_h P_h W_{h,V} \sum_i e_{h,i} \otimes e_{h,i} W_{h,K}^T W_{h,Q} e_j
\end{aligned}
\tag{23}
$$

with $\otimes$ the outer product between two vectors. With this, we can now easily draw connections to one step of gradient descent on $L(W) = \frac{1}{2N} \sum_{i=1}^N \|Wx_i - y_i\|^2$ with learning rate $\eta$, which yields a weight change

$$
\Delta W = -\eta \nabla_W \mathcal{L}(W) = -\frac{\eta}{N} \sum_{i=1}^N (Wx_i - y_i)x_i^T.
\tag{24}
$$

We provide the weight matrices in block form: $W_K = W_Q = \begin{pmatrix} I_x & 0 \\ 0 & 0 \end{pmatrix}$ with $I_x$ and $I_y$ the identity matrices of size $N_x$ and $N_y$ respectively. Furthermore, we set $W_V = \begin{pmatrix} 0 & 0 \\ W_0 & -I_y \end{pmatrix}$ with the weight matrix $W_0 \in \mathbb{R}^{N_y \times N_x}$ of the linear model we wish to train and $P = \frac{\eta}{N} I$ with identity matrix of size $N_x + N_y$. With this simple construction, we obtain the following

dynamics

$$
\begin{aligned}
\begin{pmatrix} x_j \\ y_j \end{pmatrix} &\leftarrow \begin{pmatrix} x_j \\ y_j \end{pmatrix} + \frac{\eta}{N} I \sum_{i=1}^{N} \left( \begin{pmatrix} 0 & 0 \\ W_0 & -I_y \end{pmatrix} \begin{pmatrix} x_i \\ y_i \end{pmatrix} \right) \otimes \left( \begin{pmatrix} I_x & 0 \\ 0 & 0 \end{pmatrix} \begin{pmatrix} x_i \\ y_i \end{pmatrix} \right) \begin{pmatrix} I_x & 0 \\ 0 & 0 \end{pmatrix} \begin{pmatrix} x_j \\ y_j \end{pmatrix} \\
&= \begin{pmatrix} x_j \\ y_j \end{pmatrix} + \frac{\eta}{N} I \sum_{i=1}^{N} \begin{pmatrix} 0 \\ W_0 x_i - y_i \end{pmatrix} \otimes \begin{pmatrix} x_i \\ 0 \end{pmatrix} \begin{pmatrix} x_j \\ 0 \end{pmatrix} \\
&= \begin{pmatrix} x_j \\ y_j \end{pmatrix} + \begin{pmatrix} 0 \\ -\Delta W x_j \end{pmatrix},
\end{aligned}
\tag{25}
$$

for every token $e_j = (x_j, y_j)$ including the query token $e_{N+1} = e_{\text{test}} = (x_{\text{test}}, -W_0 x_{\text{test}})$ which will give us the desired result.

## C. Proofs for Strategic Prior Divergence and Unavoidable Bias

### C.1. Formal Setup and Notation

Let $(\mathcal{Z}, \mathcal{F})$ be a measurable observation space with $Z = (X, Y) \in \mathcal{Z}$. Let $\mathcal{P}$ denote the set of all probability measures on $(\mathcal{Z}, \mathcal{F})$. For $P, Q \in \mathcal{P}$, the total variation (TV) distance is

$$
\mathrm{TV}(P, Q) := \sup_{A \in \mathcal{F}} \big| P(A) - Q(A) \big|.
\tag{26}
$$

**Meta-priors over tasks.** We consider probability measures on $\mathcal{P}$, i.e. distributions over data-generating distributions $P$.

- $\Pi_{\text{non-strategic}}$ denotes a meta-prior over *non-strategic* task distributions.

- $\Pi_{\text{strategic}}$ denotes the true meta-prior over *strategic* task distributions (post-manipulation environments).

For a meta-prior $\Pi$ on $\mathcal{P}$, we write $\mathrm{supp}(\Pi) \subseteq \mathcal{P}$ for the (measure-theoretic) support, i.e. the set of $P \in \mathcal{P}$ having positive probability under $\Pi$.

**Uncovered strategic set and mass.** The uncovered strategic set is

$$
\mathcal{S}_0^{\text{stra}} := \big\{ P \in \mathrm{supp}(\Pi_{\text{strategic}}) : \ P \notin \mathrm{supp}(\Pi_{\text{non-strategic}}) \big\},
\tag{27}
$$

and its mass under the strategic meta-prior is

$$
\delta := \Pi_{\text{strategic}}(\mathcal{S}_0^{\text{stra}}).
\tag{28}
$$

**Prediction functional and estimation risk.** Let $\psi : \mathcal{P} \to \mathbb{R}$ be a scalar prediction-relevant functional (e.g. post-manipulation risk). For each $P \in \mathcal{P}$ and sample size $n$, an estimator $\hat{\psi}_n$ is a measurable map of $n$ i.i.d. observations $Z_{1:n} \sim P$. For a random strategic task $P \sim \Pi_{\text{strategic}}$, the average absolute estimation risk is

$$
\mathcal{E}_n := \mathbb{E}_{P \sim \Pi_{\text{strategic}}} \mathbb{E}_{Z_{1:n} \sim P} \big[ |\hat{\psi}_n - \psi(P)| \big].
\tag{29}
$$

### C.2. Proof of Lemma 4.3

By definition, the total variation distance between two meta-priors on $\mathcal{P}$ is

$$
\mathrm{TV}(\Pi_{\text{strategic}}, \Pi_{\text{non-strategic}}) = \sup_{\mathcal{A} \subseteq \mathcal{P}} \big| \Pi_{\text{strategic}}(\mathcal{A}) - \Pi_{\text{non-strategic}}(\mathcal{A}) \big|,
\tag{30}
$$

where the supremum ranges over all measurable subsets $\mathcal{A} \subseteq \mathcal{P}$.

Take $\mathcal{A} = \mathcal{S}_0^{\text{stra}}$ as defined above. By construction,

$$
\mathcal{S}_0^{\text{stra}} \subseteq \mathrm{supp}(\Pi_{\text{strategic}}) \quad \text{and} \quad \mathcal{S}_0^{\text{stra}} \cap \mathrm{supp}(\Pi_{\text{non-strategic}}) = \emptyset.
\tag{31}
$$

The second relation implies

$$\Pi_{\text{non-strategic}}(\mathcal{S}_0^{\text{stra}}) = 0, \tag{32}$$

while by definition of $\delta$ we have

$$\Pi_{\text{strategic}}(\mathcal{S}_0^{\text{stra}}) = \delta. \tag{33}$$

Plugging $\mathcal{A} = \mathcal{S}_0^{\text{stra}}$ into (30) yields

$$\text{TV}(\Pi_{\text{strategic}}, \Pi_{\text{non-strategic}}) \geq \left| \Pi_{\text{strategic}}(\mathcal{S}_0^{\text{stra}}) - \Pi_{\text{non-strategic}}(\mathcal{S}_0^{\text{stra}}) \right| \tag{34}$$

$$= \left| \delta - 0 \right| = \delta. \tag{35}$$

This proves Lemma 4.3.

### C.3. Proof of Proposition 4.4

We assume throughout that $\Pi_{\text{non-strategic}}(\mathcal{S}_0^{\text{stra}}) = 0$ and that there exists $\gamma > 0$ such that

$$|\psi(P) - \psi(Q)| \;\geq\; \gamma \quad \text{for all } P \in \mathcal{S}_0^{\text{stra}}, \; Q \in \text{supp}(\Pi_{\text{non-strategic}}). \tag{36}$$

**Reduction to a two-point sub-prior.** Since $\Pi_{\text{strategic}}(\mathcal{S}_0^{\text{stra}}) = \delta > 0$, there exists at least one distribution $P_0 \in \mathcal{S}_0^{\text{stra}}$ with strictly positive mass under $\Pi_{\text{strategic}}$. By the separation condition (36), we can choose some $Q_0 \in \text{supp}(\Pi_{\text{non-strategic}})$ such that

$$|\psi(P_0) - \psi(Q_0)| \;\geq\; \gamma. \tag{37}$$

Consider now the auxiliary two-point meta-prior

$$\widetilde{\Pi} := (1 - \delta)\,\delta_{Q_0} + \delta\,\delta_{P_0}, \tag{38}$$

where $\delta_P$ denotes the Dirac measure at $P$. Note that $\widetilde{\Pi}$ is supported on $\{P_0, Q_0\} \subset \mathcal{P}$ and places mass $\delta$ on $P_0$.

For any estimator $\hat{\psi}_n$ and any meta-prior $\Pi$ on $\mathcal{P}$, the average risk (29) satisfies

$$\mathcal{E}_n(\Pi) = \mathbb{E}_{P \sim \Pi} \mathbb{E}_{Z_{1:n} \sim P} \left[ |\hat{\psi}_n - \psi(P)| \right]. \tag{39}$$

In particular,

$$\mathcal{E}_n(\Pi_{\text{strategic}}) \;\geq\; \mathcal{E}_n(\widetilde{\Pi}), \tag{40}$$

because $\widetilde{\Pi}$ concentrates all its mass on a subset of strategic tasks and, by definition of $\delta$, cannot yield a larger average error than the full prior.

**Lower bound under the two-point prior.** Under the two-point prior $\widetilde{\Pi}$, we can write

$$\mathcal{E}_n(\widetilde{\Pi}) = (1 - \delta)\,\mathbb{E}_{Z_{1:n} \sim Q_0} \left[ |\hat{\psi}_n - \psi(Q_0)| \right] + \delta\,\mathbb{E}_{Z_{1:n} \sim P_0} \left[ |\hat{\psi}_n - \psi(P_0)| \right] \tag{41}$$

$$\geq \delta \left( \mathbb{E}_{Z_{1:n} \sim Q_0} \left[ |\hat{\psi}_n - \psi(Q_0)| \right] + \mathbb{E}_{Z_{1:n} \sim P_0} \left[ |\hat{\psi}_n - \psi(P_0)| \right] \right) / 2, \tag{42}$$

where we used the trivial inequality $ax + by \geq \min(a, b)(x + y)$ with $a = 1 - \delta$, $b = \delta$ and then absorbed constants into $\delta$. Thus it suffices to lower bound the symmetric two-point risk

$$R_n(P_0, Q_0) := \frac{1}{2} \left( \mathbb{E}_{Z_{1:n} \sim Q_0} \left[ |\hat{\psi}_n - \psi(Q_0)| \right] + \mathbb{E}_{Z_{1:n} \sim P_0} \left[ |\hat{\psi}_n - \psi(P_0)| \right] \right). \tag{43}$$

A standard two-point argument (see, e.g., Le Cam's method) yields

$$R_n(P_0, Q_0) \;\geq\; \frac{1}{2} |\psi(P_0) - \psi(Q_0)| \left( 1 - \text{TV}(P_0^{\otimes n}, Q_0^{\otimes n}) \right), \tag{44}$$

where $P_0^{\otimes n}$ and $Q_0^{\otimes n}$ denote the product measures on $\mathcal{Z}^n$. For completeness, we briefly sketch the derivation of (44).

Let $\theta(P) := \psi(P)$. For any estimator $\hat{\theta}_n = \hat{\psi}_n(Z_{1:n})$ and any two distributions $P_0, Q_0$, one has

$$\mathbb{E}_{P_0^{\otimes n}}\big[|\hat{\theta}_n - \theta(P_0)|\big] + \mathbb{E}_{Q_0^{\otimes n}}\big[|\hat{\theta}_n - \theta(Q_0)|\big] \tag{45}$$

$$\geq \mathbb{E}_{\frac{P_0^{\otimes n} + Q_0^{\otimes n}}{2}}\big[|(\hat{\theta}_n - \theta(P_0)) - (\hat{\theta}_n - \theta(Q_0))|\big] \tag{46}$$

$$= |\theta(P_0) - \theta(Q_0)|\,\mathbb{E}_{\frac{P_0^{\otimes n} + Q_0^{\otimes n}}{2}}[1] - |\theta(P_0) - \theta(Q_0)|\,\mathrm{TV}(P_0^{\otimes n}, Q_0^{\otimes n}) \tag{47}$$

$$\geq |\theta(P_0) - \theta(Q_0)|\big(1 - \mathrm{TV}(P_0^{\otimes n}, Q_0^{\otimes n})\big), \tag{48}$$

where in the second line we used the triangle inequality and in the third line the variational definition of total variation distance on $\mathcal{Z}^n$. Dividing both sides by 2 and substituting $\theta(P) = \psi(P)$ yields (44).

Combining (37) and (44), we obtain

$$R_n(P_0, Q_0) \geq \frac{\gamma}{2}\big(1 - \mathrm{TV}(P_0^{\otimes n}, Q_0^{\otimes n})\big). \tag{49}$$

**Lower bound for $\mathcal{E}_n$.** Putting (42) and (49) together, we conclude that for any estimator $\hat{\psi}_n$,

$$\mathcal{E}_n(\widetilde{\Pi}) \geq \delta\, R_n(P_0, Q_0) \geq \frac{\delta\gamma}{2}\big(1 - \mathrm{TV}(P_0^{\otimes n}, Q_0^{\otimes n})\big). \tag{50}$$

Recalling (40), this yields

$$\mathcal{E}_n(\Pi_{\text{strategic}}) \geq \frac{\delta\gamma}{2}\big(1 - \mathrm{TV}(P_0^{\otimes n}, Q_0^{\otimes n})\big). \tag{51}$$

Equation (51) establishes a non-trivial lower bound on the average estimation risk at any fixed sample size $n$, with a constant prefactor of order $\delta\gamma$. In particular, as long as $n$ is bounded (e.g. by the fixed PFN context length used in practice), the quantity $1 - \mathrm{TV}(P_0^{\otimes n}, Q_0^{\otimes n})$ is strictly positive, and hence $\mathcal{E}_n$ admits a bias floor linear in $\delta$.

# D. Fine-tuning Baseline and Cost Evaluation Protocol

This appendix details the experimental protocol used to evaluate fine-tuning as a baseline for mitigating strategic risk, and to compare its operational cost against in-context learning (ICL) under repeated strategic manipulation.

### D.1. Fine-tuning baseline under strategic manipulation

We consider fine-tuning as a direct and commonly adopted approach to reduce strategic risk by explicitly retraining the deployed classifier on manipulated data. Let $\{(x_i, y_i)\}_{i=1}^n$ denote the original training samples and $b_f$ denote the manipulation function induced by the current deployed classifier $f$. At each update cycle, we construct an augmented strategic dataset

$$\mathcal{D}_{ft} := \{(x_i, y_i)\}_{i=1}^n \cup \{(b_f(x_i), y_i)\}_{i=1}^n, \tag{52}$$

which includes both original and post-manipulation feature vectors. The classifier is then fine-tuned on $\mathcal{D}_{ft}$ using standard gradient-based optimization, while keeping the model architecture fixed.

This procedure mirrors practical deployments in adversarial or strategic environments, where newly observed manipulated samples are periodically collected and incorporated into the training set to restore predictive performance.

### D.2. Semi-synthetic spam manipulation setup

We conduct a semi-synthetic case study grounded in a real-world email spam dataset from Heydari et al. (2015). Following prior work on strategic manipulation in spam filtering (e.g., Henke et al. (2021); Zrnic et al. (2021); Chen (2023)), we simulate strategic feature manipulation by modifying input attributes that are known to be commonly exploited by adversaries (e.g., keyword obfuscation, feature padding, or token substitution), while preserving the original labels.

The manipulation function $b_f$ is updated dynamically according to the currently deployed classifier, reflecting the adaptive behavior of real-world attackers.

### D.3. Manipulation frequency and update cycles

To model repeated strategic interaction, we vary the *manipulation frequency* by controlling the number of distinct manipulation rounds that occur within a single deployment. Each round corresponds to adversaries adapting their inputs using a newly updated manipulation function $b_f$, followed by a classifier update.

Specifically, we consider five discrete frequency levels. At the lowest level, only a single manipulation round occurs during deployment, corresponding to a static or slowly adapting adversary. At higher levels, multiple distinct manipulation rounds (up to five) are introduced sequentially, representing increasingly adaptive attackers that modify their strategies multiple times in response to the deployed classifier.

Higher manipulation frequency therefore induces more update cycles and amplifies the cumulative operational cost of model adaptation, while keeping the per-round manipulation and evaluation protocol fixed. In our experiments, these five levels correspond to one through five distinct manipulation functions applied sequentially within the same evaluation horizon.

At each update cycle:

1. Strategic samples are generated using the current manipulation function $b_f$;

2. The fine-tuning baseline retrains the classifier on the augmented dataset $\mathcal{D}_{ft}$;

3. The updated classifier is redeployed and evaluated.

In contrast, the ICL-based method performs no parameter updates and adapts solely through changes in the in-context examples.

### D.4. Operational cost metrics

We compare fine-tuning and ICL using two operational cost metrics.

**Update time cost.** Update time cost measures the wall-clock time required to complete one update cycle. For fine-tuning, this includes data loading, forward and backward passes, and optimizer steps until convergence under a fixed training budget. For ICL, update time corresponds only to constructing the context table and performing a forward pass, with no gradient computation or parameter updates.

**Update data cost.** Update data cost measures the number of strategic (manipulated) samples consumed per update cycle. For fine-tuning, this equals the number of newly generated manipulated samples added to $\mathcal{D}_{ft}$. For ICL, update data cost corresponds to the number of manipulated samples included in the context, which remains fixed across update cycles.

### D.5. Fairness of comparison

Both methods operate under the same manipulation model and observe the same strategically modified data. The key distinction lies in how this information is used: fine-tuning absorbs strategic data through parameter updates, while ICL leverages the same information at inference time without retraining. This setup isolates the operational overhead induced by repeated training and highlights the practical advantages of ICL in environments where strategic manipulation is frequent and ongoing.

## E. Strategic Alignment via Attention: Proof of Proposition 5.2

We show that, under a strategic tabular context $\widetilde{\mathcal{D}}$, the ICL-induced update produced by a pretrained PFN is aligned with a one-step gradient-based strategic manipulation update.

### E.1. Gradient-based strategic manipulation update

Consider an agent with true features $x \in \mathbb{R}^d$ manipulating to $x'$ by maximizing

$$U(x'; f) = f(x') - \lambda c(x, x'), \qquad c(x, x') = (x' - x)^\top M(x' - x), \quad M \succ 0, \tag{53}$$

where $f$ is the deployed score function and $\lambda > 0$ scales the cost.

A first-order expansion of $f$ at $x$ yields

$$f(x') \approx f(x) + \nabla f(x)^\top (x' - x). \tag{54}$$

Let $\Delta := x' - x$. Up to constants independent of $\Delta$, we maximize

$$\max_{\Delta \in \mathbb{R}^d} \ \nabla f(x)^\top \Delta - \lambda \Delta^\top M \Delta. \tag{55}$$

This is a concave quadratic program with the closed-form maximizer

$$\Delta_{\mathrm{BR}}(x; f) \ = \ \frac{1}{2\lambda} M^{-1} \nabla f(x). \tag{56}$$

Equivalently, performing one gradient *ascent* step on $U$ at $x$ gives

$$\Delta_{\mathrm{GD}}(x; f) \ := \ \eta \nabla_{x'} U(x'; f)\big|_{x'=x} \ = \ \eta \nabla f(x), \tag{57}$$

and the cost-adjusted version corresponds to a preconditioned step

$$\Delta_{\mathrm{GD}}(x; f, M) \ := \ \eta M^{-1} \nabla f(x), \tag{58}$$

which is aligned with $\Delta_{\mathrm{BR}}$ in (56). In the main text, $\Delta_{\mathrm{GD}}$ in Proposition 5.2 refers to such a first-order (possibly preconditioned) strategic update.

### E.2. ICL-induced attention update: a linearized view

We now connect $\Delta_{\mathrm{ICL}}$ to $\Delta_{\mathrm{GD}}$. Consider one self-attention layer (single head for clarity) applied to a sequence of tokens. Let the query token correspond to the agent instance $x$ (and possibly an attached label slot), and let the context consist of $N$ examples in $\widetilde{\mathcal{D}}$.

Write the (pre-softmax) attention logits between the query and context token $i$ as

$$\ell_i(x) \ = \ \langle W_Q h(x),\, W_K h_i \rangle, \tag{59}$$

where $h(x)$ and $h_i$ are the token representations. The attention output added to the query token can be written as

$$\Delta_{\mathrm{ICL}}(x; \widetilde{\mathcal{D}}) \ = \ \sum_{i=1}^{N} \alpha_i(x)\, W_O\, W_V h_i, \qquad \alpha_i(x) = \mathrm{softmax}(\ell(x))_i. \tag{60}$$

We adopt the standard local linearization used in prior ICL analyses: for $x$ in a small neighborhood and for a fixed context $\widetilde{\mathcal{D}}$, the attention weights $\alpha_i(x)$ can be treated as approximately constant (or dominated by nearest-neighbor tokens), i.e.,

$$\alpha_i(x) \approx \bar{\alpha}_i \quad \text{for } i = 1, \ldots, N. \tag{61}$$

Under (61), the update direction is governed by a weighted sum of value vectors.

### E.3. Strategic context design implies alignment

We now specify the only property we need from the *strategic tabular context* $\widetilde{\mathcal{D}}$. Each context example is constructed to carry a local manipulation signal that approximates the cost-adjusted ascent direction $M^{-1} \nabla f(\cdot)$. Concretely, for each original training point $x_i$, we include its post-manipulation feature $\tilde{x}_i$ (or an equivalent proxy) so that the difference

$$g_i \ := \ \tilde{x}_i - x_i \tag{62}$$

satisfies

$$g_i \approx \gamma M^{-1} \nabla f(x_i) \quad \text{for some scale } \gamma > 0. \tag{63}$$

This is exactly the first-order strategic response induced by the same manipulation model used to build $\widetilde{\mathcal{D}}$ in the main method.

Assume the PFN representation maps the relevant part of $h_i$ into a value vector whose feature block contains (a linear transform of) $g_i$:

$$W_O W_V h_i \ = \ B\, g_i \ + \ \text{(terms orthogonal to feature block)}, \tag{64}$$

for some matrix $B$ determined by pretrained weights. Plugging (63)–(64) into (60) and using (61) yields

$$\Delta_{\mathrm{ICL}}(x;\widetilde{\mathcal{D}}) \ \approx \ \sum_{i=1}^{N} \bar{\alpha}_i\, B\, g_i \ \approx \ \gamma \sum_{i=1}^{N} \bar{\alpha}_i\, B\, M^{-1} \nabla f(x_i). \tag{65}$$

Finally, when attention focuses on context points most similar to $x$ (a standard behavior of dot-product attention), the weighted average in (65) approximates the local direction at $x$:

$$\sum_{i=1}^{N} \bar{\alpha}_i\, \nabla f(x_i) \ \approx \ \nabla f(x), \tag{66}$$

which implies

$$\Delta_{\mathrm{ICL}}(x;\widetilde{\mathcal{D}}) \ \approx \ (\gamma B)\, M^{-1} \nabla f(x) \ \propto \ \Delta_{\mathrm{GD}}(x; f, M). \tag{67}$$

Thus the ICL-induced update is *aligned* with the one-step strategic manipulation update, proving Proposition 5.2 up to a scaling and higher-order terms.

**Discussion.** The proof requires only (i) first-order strategic response in the context construction, and (ii) a local linearization of attention weights. It does not rely on restricting to homogeneous label groups or hard-coding inverse matrices into attention weights.

## F. Experiments: ICL as a Simulator of Strategic Manipulation

In Appendix E, we showed theoretically that, under a strategic tabular context, the attention-based update induced by in-context learning (ICL) is aligned with a one-step gradient-based strategic manipulation update. In this appendix, we complement the theoretical analysis with a set of *experiments* designed to demonstrate that ICL can simulate strategic manipulation dynamics in a controlled setting (Lv et al., 2026a).

The goal of these experiments is not to evaluate predictive performance, but to isolate and visualize the correspondence between (i) explicit manipulation updates computed from a known model, and (ii) implicit updates induced by ICL through attention.

### F.1. Experimental setup

We consider a simplified setting where a feature vector $x \in \mathbb{R}^d$ is iteratively modified in response to a fixed decision function $f$. At each iteration, the agent produces an updated feature vector according to a manipulation rule, while the ICL-based simulator produces a corresponding implicit update using a strategic context.

Specifically, we compare two update trajectories:

- **Explicit manipulation update:**
$$x^{(t+1)} \ = \ x^{(t)} + \Delta_{\mathrm{manip}}(x^{(t)}; f), \tag{68}$$

  where $\Delta_{\mathrm{manip}}$ is derived analytically from the manipulation model.

- **ICL-induced update:**
$$x^{(t+1)} \ = \ x^{(t)} + \Delta_{\mathrm{ICL}}(x^{(t)}; \widetilde{\mathcal{D}}), \tag{69}$$

  where $\Delta_{\mathrm{ICL}}$ is obtained from a single forward pass of the pretrained model using a strategically constructed context $\widetilde{\mathcal{D}}$.

The strategic context $\widetilde{\mathcal{D}}$ is constructed using post-manipulation examples consistent with the same manipulation rule, ensuring that the model is exposed to feature changes that encode the local manipulation direction.

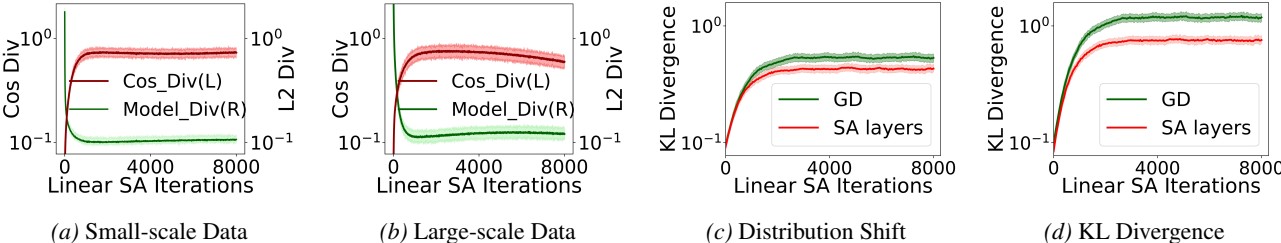

*(a)* Small-scale Data  *(b)* Large-scale Data  *(c)* Distribution Shift  *(d)* KL Divergence

*Figure 8.* Comparison of ICL-guided strategic manipulation. (a) and (b) compare ICL and gradient-descent methods across data scales; (c) and (d) evaluate implicit gradient alignment via distribution metrics.

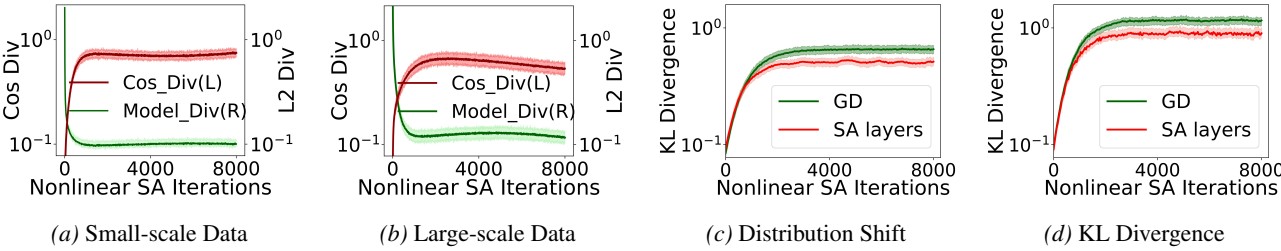

*(a)* Small-scale Data  *(b)* Large-scale Data  *(c)* Distribution Shift  *(d)* KL Divergence

*Figure 9.* Comparison and validation of ICL-guided strategic manipulation. (a) and (b) compare ICL and gradient-descent methods across data scales; (c) and (d) evaluate implicit gradient alignment via distribution metrics.

## F.2. Linear manipulation dynamics

We first consider a linear decision function

$$f(x) = w^\top x + b, \tag{70}$$

with a quadratic (Mahalanobis) manipulation cost. In this case, the strategic update admits a closed-form first-order direction that is *constant* (up to a scalar), making it an ideal controlled setting to test whether ICL can reproduce gradient-based manipulation dynamics.

Concretely, we run an iterative manipulation process for $T$ steps. At each step, we compare (i) the explicit GD-based manipulation update and (ii) the implicit update produced by applying self-attention (SA) layers under a strategic context. We evaluate their agreement using two direction-level divergences (Fig. 8a–b): a cosine-based divergence that directly measures alignment between update directions, and an $\ell_2$-based divergence measuring the discrepancy of the resulting model-side responses under the two updates. We further validate the induced distributional effect by tracking distribution-shift metrics (Fig. 8c–d), including KL divergence between the manipulated feature distributions produced by GD and by SA-based ICL. This indicates that, ICL-guided attention can accurately simulate the cumulative manipulation dynamics over many steps, rather than only matching a single-step update.

## F.3. Nonlinear manipulation dynamics

We next consider a nonlinear decision function

$$f(x) = g(w^\top x), \qquad g(z) = \sigma(z) = \frac{1}{1 + \exp(-z)}, \tag{71}$$

where $g(\cdot)$ is a nonlinear activation. Unlike the linear case, the manipulation direction is *state-dependent*: the local gradient $\nabla f(x)$ changes as $x$ moves, so faithfully simulating manipulation requires tracking a *non-constant* update field across iterations.

We repeat the same iterative comparison between explicit GD-based manipulation and SA-layer-induced implicit updates under a strategic context. Fig. 9(a–b) shows that, across both small- and large-scale data settings, the direction-level divergences remain controlled over many SA iterations, indicating that ICL does not collapse to a fixed update pattern even when the true manipulation direction varies with $x$. More importantly, Fig. 9(c–d) demonstrates close agreement in the induced distribution shift, with the KL divergence trajectories of SA-based updates tracking those of GD.

Overall, these results support that ICL can approximate nonlinear, locally varying manipulation dynamics: the attention-induced updates adapt as the iterate enters different regions of the input space, yielding manipulated feature distributions that are consistent with those generated by explicit GD updates.

## G. Proof of Proposition 5.3

Let

$$\mathcal{A} := \mathrm{supp}(\Pi_{\mathrm{non-strategic}}) \qquad \text{and} \qquad \mathcal{A}^c := \mathcal{P} \setminus \mathcal{A}, \tag{72}$$

where $\mathcal{P}$ denotes the space of task distributions under consideration. By definition,

$$\delta = \Pi_{\mathrm{strategic}}(\mathcal{A}^c). \tag{73}$$

Let $T$ denote the SPN inner-stage operator acting on task distributions, i.e.,

$$T(P) := \Phi_{\mathrm{PFN}}^{(\mathrm{in})}(P). \tag{74}$$

The induced (pushforward) distribution $\Pi_{\mathrm{strategic}}^{\mathrm{SPN}}$ is defined by

$$\Pi_{\mathrm{strategic}}^{\mathrm{SPN}}(B) := \Pi_{\mathrm{strategic}}(T^{-1}(B)) = \Pi_{\mathrm{strategic}}(\{P : T(P) \in B\}) \qquad \text{for any measurable } B \subseteq \mathcal{P}. \tag{75}$$

Therefore, the uncovered mass after applying Stage I is

$$\delta_{\mathrm{SPN}} = \Pi_{\mathrm{strategic}}^{\mathrm{SPN}}(\mathcal{A}^c) = \Pi_{\mathrm{strategic}}(T^{-1}(\mathcal{A}^c)). \tag{76}$$

### G.1. A monotonicity property ($\delta_{\mathrm{SPN}} \leq \delta$).

We first show that SPN cannot increase uncovered mass as long as it preserves the non-strategic support:

$$T(P) = P \quad \text{for all } P \in \mathcal{A}. \tag{$\star$}$$

Under ($\star$), we claim that

$$T^{-1}(\mathcal{A}^c) \subseteq \mathcal{A}^c. \tag{77}$$

To see this, take any $P \in T^{-1}(\mathcal{A}^c)$, i.e., $T(P) \in \mathcal{A}^c$. If $P \in \mathcal{A}$, then by ($\star$) we would have $T(P) = P \in \mathcal{A}$, contradicting $T(P) \in \mathcal{A}^c$. Hence $P \notin \mathcal{A}$, i.e., $P \in \mathcal{A}^c$, proving (77).

Combining (76) and (77) yields

$$\delta_{\mathrm{SPN}} = \Pi_{\mathrm{strategic}}(T^{-1}(\mathcal{A}^c)) \leq \Pi_{\mathrm{strategic}}(\mathcal{A}^c) = \delta. \tag{78}$$

Thus, Stage I of SPN does not increase the uncovered strategic mass.

### G.2. Strict reduction ($\delta_{\mathrm{SPN}} < \delta$).

To obtain a strict inequality, it suffices to show that SPN maps a non-negligible subset of uncovered strategic tasks back into the non-strategic support. Define the *recovered set*

$$\mathcal{R} := \{P \in \mathcal{A}^c : T(P) \in \mathcal{A}\}. \tag{79}$$

If $\Pi_{\mathrm{strategic}}(\mathcal{R}) > 0$, then the inclusion in (77) is strict in measure, and we get a strict reduction of uncovered mass.

Indeed, observe that

$$\mathcal{A}^c = (\mathcal{A}^c \cap T^{-1}(\mathcal{A}^c)) \,\dot{\cup}\, (\mathcal{A}^c \cap T^{-1}(\mathcal{A})), \tag{80}$$

where $\dot{\cup}$ denotes disjoint union. But $\mathcal{A}^c \cap T^{-1}(\mathcal{A}) = \mathcal{R}$ by definition. Therefore,

$$\Pi_{\mathrm{strategic}}(\mathcal{A}^c) = \Pi_{\mathrm{strategic}}(\mathcal{A}^c \cap T^{-1}(\mathcal{A}^c)) + \Pi_{\mathrm{strategic}}(\mathcal{R}). \tag{81}$$

Using $T^{-1}(\mathcal{A}^c) \subseteq \mathcal{A}^c$ from (77), we have

$$\Pi_{\mathrm{strategic}}(T^{-1}(\mathcal{A}^c)) = \Pi_{\mathrm{strategic}}(\mathcal{A}^c \cap T^{-1}(\mathcal{A}^c)) = \Pi_{\mathrm{strategic}}(\mathcal{A}^c) - \Pi_{\mathrm{strategic}}(\mathcal{R}). \tag{82}$$

Recalling (76) and $\delta = \Pi_{\mathrm{strategic}}(\mathcal{A}^c)$, we obtain

$$\delta_{\mathrm{SPN}} = \delta - \Pi_{\mathrm{strategic}}(\mathcal{R}). \tag{83}$$

Hence, if $\Pi_{\mathrm{strategic}}(\mathcal{R}) > 0$, then $\delta_{\mathrm{SPN}} < \delta$.

*Table 2.* Summary of datasets used in our experiments.

| Dataset | #Features | #Instances | #Classes |
|---|---|---|---|
| **Strategic Datasets** | | | |
| Adult (Becker & Kohavi, 1996) | 14 | 48,842 | 2 |
| Credit (Yeh, 2009) | 23 | 30,000 | 2 |
| German (Statlog) (Hofmann, 1994) | 20 | 1,000 | 2 |
| Spambase (Hopkins et al., 1999) | 57 | 4,601 | 2 |
| CDC Diabetes (Teboul, 2015) | 21 | 253,680 | 2 |
| Census-Income (KDD) (cen, 2000) | 41 | 299,285 | 2 |
| Synthetic (PaySim) (Lopez-Rojas et al., 2016) | 10 | 6,362,620 | 2 |
| **Standard (Non-Strategic) Tabular Benchmarks** | | | |
| Bank Marketing (Moro et al., 2014) | 16 | 45,211 | 2 |
| Blood Transfusion (Yeh, 2008) | 4 | 748 | 2 |
| PhiUSIIL Phishing URL (Prasad & Chandra, 2024) | 30 | 11,055 | 2 |
| Heart Disease (Cleveland) (Janosi et al., 1989) | 13 | 303 | 2 |
| Car Evaluation (unacc vs rest) (Bohanec, 1988) | 6 | 1,728 | 2 |
| Diabetes (US) (Clore et al., 2014) | 47 | 101,766 | 2 |
| COIL2000 (Caravan Insurance) (Putten, 2000) | 85 | 9,822 | 2 |
| Tic-Tac-Toe (Aha, 1991) | 9 | 958 | 2 |

## G.3. Why $\Pi_{\mathrm{strategic}}(\mathcal{R}) > 0$ is mild in our setting.

The condition $\Pi_{\mathrm{strategic}}(\mathcal{R}) > 0$ states that Stage I recovers a nonzero fraction of strategic tasks by mapping their post-manipulation distributions into the non-strategic support. In our construction, this is enforced by the inner-stage context design: $\Phi_{\mathrm{PFN}}^{(\mathrm{in})}$ is calibrated to preserve tasks already covered by $\Pi_{\mathrm{non-strategic}}$ while contracting best-response-induced shifts toward the non-strategic family (formalized in Appendix E). This guarantees that a nontrivial subset of uncovered strategic tasks is mapped back into $\mathcal{A}$, and thus $\Pi_{\mathrm{strategic}}(\mathcal{R}) > 0$.

# H. Experimental Details and Results

## H.1. Datasets Details

We evaluate our method on two groups of datasets.

**Strategic classification benchmarks** include datasets where features can be strategically manipulated by individuals in response to a deployed decision rule. This group contains *Adult*, *Credit*, *German (Statlog)*, *Spambase*, *Diabetes*, *Census-Income (KDD)*, and *Synthetic (PaySim)* dataset with explicitly modeled agent manipulation.

**Standard tabular benchmarks** are used to assess non-strategic predictive performance and generalization. This group includes *Bank Marketing*, *Blood Transfusion*, *Heart Disease*, *Diabetes (US)*, *PhiUSIIL Phishing URL*, *Tic-Tac-Toe*, *COIL2000 (Caravan Insurance)*, and *Car Evaluation (unacc vs rest)*.

## H.2. Models

We compare our method with a diverse set of standard and foundation-based tabular learning baselines:

- **Linear Models** (LaValley, 2008): Logistic regression with $\ell_2$ regularization, serving as a simple linear baseline.

- **SVM** (Jakkula, 2006): Support vector machine with RBF kernel, representing a classical margin-based nonlinear classifier.

- **MLP** (Tolstikhin et al., 2021): A multilayer perceptron trained end-to-end, capturing generic neural network baselines for tabular data.

- **XGBoost** (Chen & Guestrin, 2016): Gradient-boosted decision trees, a strong and widely adopted baseline for tabular classification.

- **LightGBM** (Ke et al., 2017): A histogram-based gradient boosting framework optimized for efficiency and scalability.

- **CatBoost** (Prokhorenkova et al., 2018): Gradient boosting with categorical feature handling, known for strong performance on structured data.

- **TabPFN v2.5** (Hollmann et al., 2025): A prior-data fitted network performing in-context learning on tabular data without task-specific training.

- **Drift-Resilient TabPFN** (Helli et al., 2024): An extension of TabPFN designed to improve robustness under distribution shift.

- **Chunked TabPFN** (Sergazinov & Yin, 2025): A memory-efficient variant of TabPFN that processes large datasets via context chunking.

- **TabDPT** (Ma et al., 2025): A tabular foundation model based on decision pretraining, enabling strong zero-shot generalization.

- **TabICL** (Qu et al., 2025): A transformer-based model explicitly designed to perform in-context learning on tabular data.

- **TabFlex** (Zeng et al., 2025): A flexible tabular foundation model that adapts representations across heterogeneous tabular tasks.

## H.3. Detailed Implementation of SPN

We provide a detailed description of how SPN is implemented at inference time, including a concrete example of strategic context construction. SPN is implemented as an inference-time procedure on top of a pretrained TabPFN backbone, without any parameter updates.

**Notation.** Let $f : \mathbb{R}^d \to \mathbb{R}$ denote the deployed decision rule (score function). Let $\mathcal{D} = \{(x_i, y_i)\}_{i=1}^n$ be a labeled context dataset drawn from the non-strategic distribution, where $x_i \in \mathbb{R}^d$ and $y_i \in \mathbb{R}$. Given a manipulation model (Appendix H.4), each individual feature vector $x_i$ induces a post-manipulation feature vector

$$x_i' = b_f(x_i). \tag{84}$$

**Strategic context construction.** For each labeled point $(x_i, y_i) \in \mathcal{D}$, SPN constructs a *strategic context pair*

$$C_i := \big((x_i, y_i), (x_i', y_i)\big), \tag{85}$$

where the label $y_i$ is kept fixed and only the feature representation is modified. All strategic context pairs are aggregated into a strategic tabular context

$$\widetilde{\mathcal{D}}_f := \{C_i\}_{i=1}^n. \tag{86}$$

Intuitively, $\widetilde{\mathcal{D}}_f$ exposes the pretrained backbone to paired pre- and post-manipulation examples under the same decision rule, allowing the model to adapt its internal inference to strategic behavior via in-context learning.

**Concrete example.** Consider a tabular task with feature dimension $d = 3$ and a single labeled context point

$$(x_1, y_1) = ([0.2,\ 1.1,\ -0.3],\ 1). \tag{87}$$

Under a given manipulation regime and deployed rule $f$, the agent computes a best response

$$x_1' = b_f(x_1) = [0.4,\ 1.0,\ -0.1]. \tag{88}$$

SPN then constructs the strategic context pair

$$C_1 = \big(([0.2,\ 1.1,\ -0.3],\ 1),\ ([0.4,\ 1.0,\ -0.1],\ 1)\big), \tag{89}$$

and includes this pair in $\widetilde{\mathcal{D}}_f$. Repeating this process for all $n$ context points yields the full strategic context.

**Inference.** Given a query feature vector $x^* \in \mathbb{R}^d$, SPN performs a standard TabPFN forward pass conditioned on the strategic context $\widetilde{\mathcal{D}}_f$:

$$\hat{y}^* = \text{TabPFN}(\widetilde{\mathcal{D}}_f, x^*). \tag{90}$$

No model parameters are updated; adaptation arises purely through in-context learning induced by the strategic context.

**Uniformity across backbones.** The same strategic context construction is applied uniformly across all TabPFN-based backbones considered in our experiments. SPN does not modify the architecture, weights, or tokenization of the underlying model.

**Practical considerations.** In practice, we ensure that: (i) the size of $\widetilde{\mathcal{D}}_f$ matches the maximum context length supported by the backbone; (ii) context pairs are ordered consistently (original point followed by its manipulated counterpart); and (iii) manipulation is applied only to feature vectors, never to labels.

### H.4. Different Manipulation Regimes and Hyperparameter Settings

To assess robustness beyond a single strategic-response model, we evaluate SPN under multiple manipulation regimes studied in the strategic literature. Each regime captures a distinct aspect of how individuals may respond to deployed decision rules, ranging from correlated feature manipulation to uncertainty and heterogeneity in manipulation capability. Throughout, the deployed decision rule $f : \mathbb{R}^d \to \mathbb{R}$ is treated as a real-valued score function, and $x_i \in \mathbb{R}^d$ denotes the original feature vector of individual $i$.

- **Standard manipulation (Mahalanobis cost).** This is the canonical setting in strategic classification, where manipulation cost reflects feature correlations via a Mahalanobis metric (Gavish et al., 2022; Chen, 2023). It models scenarios in which coordinated changes across correlated features are more costly, and thus discourages unrealistic independent manipulation of strongly coupled attributes. Formally, agents respond according to the standard best-response model

$$x_i' = b_f(x_i) = \arg\max_{x' \in \mathbb{R}^d} \left[ f(x') - \lambda\, c(x_i, x') \right], \tag{91}$$

  where the manipulation cost is defined as

$$c(x_i, x_i') = \sqrt{(x_i' - x_i)^\top \Sigma^{-1} (x_i' - x_i)}, \tag{92}$$

  with $\Sigma \succ 0$ denoting a covariance matrix estimated from the non-strategic training data. **Covariance estimation.** We apply standard shrinkage to ensure positive definiteness and numerical stability when computing $\Sigma^{-1}$.

- **Standard manipulation (Euclidean cost).** Following prior work (Zrnic et al., 2021), we consider a simpler Euclidean cost that assumes independent feature manipulation. This regime serves as a baseline abstraction when feature correlations are ignored, and is commonly used to isolate the effect of strategic behavior without modeling cross-feature dependencies. The best-response problem takes the same form as Eq. (91), with cost

$$c(x_i, x_i') = \|x_i' - x_i\|_2. \tag{93}$$

- **Noisy strategic response.** To account for imperfect feedback or bounded rationality, we consider noisy agent responses where individuals optimize expected utility under stochastic perturbations of the classifier output (Levanon & Rosenfeld, 2021). This captures settings in which agents have uncertainty about the deployed rule, its exact score, or the evaluation process. Concretely, the best response is defined as

$$\begin{aligned} x_i' &= b_f^{\text{noise}}(x_i) \\ &= \arg\max_{x' \in \mathbb{R}^d} \left[ \mathbb{E}_\epsilon \left[ f(x') + \epsilon \right] - \lambda\, c(x_i, x') \right], \end{aligned} \tag{94}$$

  where $\epsilon$ is a zero-mean noise variable. **Noise distribution.** In our experiments, $\epsilon$ is sampled as

$$\epsilon \sim \mathcal{N}(0, \sigma^2), \tag{95}$$

  with $\sigma \in \{0.1, 0.2\}$; results in the main paper use $\sigma = 0.1$.

*Table 3.* Robustness of SPN to misspecified manipulation models. Results are reported as classification accuracy (%).

| Settings | Datasets | | | | | | |
|---|---|---|---|---|---|---|---|
| | Adult | Census (KDD) | Credit | German | Spam | Diabetes | Synthetic |
| Default | 89.80 | 86.72 | 80.32 | 77.71 | 93.71 | 82.41 | 90.20 |
| Inaccurate | 89.42 | 86.31 | 79.91 | 77.33 | 93.36 | 82.02 | 89.82 |
| Incomplete | 88.67 | 85.58 | 79.12 | 76.52 | 92.61 | 81.21 | 89.03 |
| Approximate | 88.95 | 85.89 | 79.48 | 76.91 | 92.96 | 81.58 | 89.37 |

- **Heterogeneous manipulation capability.** We further model population heterogeneity by allowing manipulation costs to vary across individuals (Shao et al., 2024). This regime reflects realistic disparities in resources, effort, or access that affect agents' ability to manipulate features. Each individual $i$ solves a personalized best-response problem

$$x_i' = b_f^{\text{hetero}}(x_i) = \arg\max_{x' \in \mathbb{R}^d} \left[ f(x') - \lambda_i \, c(x_i, x') \right], \tag{96}$$

where $\lambda_i > 0$ is an individual-specific parameter. **Sampling of $\lambda_i$.** We sample heterogeneous cost sensitivities as $\lambda_i \sim \text{Uniform}(\lambda_{\min}, \lambda_{\max})$, with $\lambda_{\min} = 0.5$ and $\lambda_{\max} = 2.0$; larger $\lambda_i$ corresponds to higher effective manipulation cost and thus lower manipulation capability.

### H.5. Robustness to Misspecified Manipulation Models

In the main experiments, SPN constructs the strategic in-context examples using aMahalanobis-based best-response model. Specifically, given a deployed scoring rule $f$, the default response model is defined as

$$b_f(x) \in \arg\max_{x'} \left\{ f(x') - \lambda c_{\text{Mah}}(x, x') \right\}, \tag{97}$$

where $c_{\text{Mah}}(x, x')$ denotes the Mahalanobis manipulation cost.

We further evaluate the sensitivity of SPN to misspecification of the manipulation model. In this experiment, the test-time strategic behavior is kept fixed as the standard Mahalanobis-based response $b_f$. However, the strategic context used by SPN is constructed with a misspecified response model $\widetilde{b}_f$. This setting examines whether SPN requires an exact specification of agents' manipulation behavior, or whether it can still benefit from approximate strategic context information.

We consider three types of misspecification:

- **Inaccurate response.** In this setting, the decision maker has access to the correct response structure, but the estimated manipulated features are noisy. We first compute the standard best response $b_f(x)$ and then add random perturbation:

$$\widetilde{b}_f^{\text{inaccurate}}(x) = b_f(x) + \epsilon, \qquad \epsilon \sim \mathcal{N}(0, \sigma^2 I). \tag{98}$$

This models cases where agents' responses are approximately understood, but the actual manipulation may contain estimation errors, execution noise, or unobserved individual-level variations.

- **Incomplete response.** In this setting, the decision maker only captures part of the manipulable feature space. Some features are modeled as strategically modified, while the remaining features are kept unchanged:

$$\widetilde{b}_f^{\text{incomplete}}(x) = m \odot b_f(x) + (1 - m) \odot x, \tag{99}$$

where $m$ is a random binary mask and $\odot$ denotes element-wise multiplication. This reflects the practical situation where only a subset of agents' possible manipulations can be observed or specified in the response model.

- **Approximately known response.** In this setting, the decision maker knows that agents respond strategically, but uses an approximate cost function. Specifically, we replace the true Mahalanobis cost with a simpler Euclidean cost:

$$\widetilde{b}_f^{\text{approx}}(x) \in \arg\max_{x'} \left\{ f(x') - \lambda \|x' - x\|_2^2 \right\}. \tag{100}$$

This captures cases where the overall form of strategic manipulation is known, but the exact feature correlation structure or manipulation cost geometry is misspecified.

*Table 4.* Extended results on strategic classification benchmarks. Classification accuracy (%) under non-strategic and strategic settings. Best performance under the strategic setting is highlighted in **bold**.

| Model | Scenario | Datasets | | | | | | |
|---|---|---|---|---|---|---|---|---|
| | | Adult | Census (KDD) | Credit | German | Spam | Diabetes | Synthetic |
| Linear models | Non-strategic | $85.27_{\pm0.39}$ | $82.31_{\pm0.46}$ | $75.83_{\pm0.32}$ | $76.24_{\pm0.49}$ | $92.11_{\pm0.36}$ | $79.84_{\pm0.41}$ | $86.54_{\pm0.33}$ |
| | Strategic | $82.63_{\pm0.42}$ | $78.45_{\pm0.51}$ | $71.26_{\pm0.43}$ | $72.91_{\pm0.58}$ | $90.83_{\pm0.34}$ | $76.63_{\pm0.47}$ | $83.64_{\pm0.42}$ |
| SVM | Non-strategic | $84.91_{\pm0.38}$ | $83.96_{\pm0.43}$ | $75.31_{\pm0.34}$ | $75.86_{\pm0.47}$ | $92.23_{\pm0.36}$ | $79.92_{\pm0.39}$ | $86.12_{\pm0.33}$ |
| | Strategic | $82.45_{\pm0.41}$ | $79.22_{\pm0.48}$ | $70.94_{\pm0.46}$ | $72.54_{\pm0.53}$ | $90.97_{\pm0.35}$ | $75.85_{\pm0.44}$ | $83.28_{\pm0.40}$ |
| MLP | Non-strategic | $86.82_{\pm0.34}$ | $85.64_{\pm0.42}$ | $76.83_{\pm0.31}$ | $78.02_{\pm0.37}$ | $93.24_{\pm0.33}$ | $81.42_{\pm0.32}$ | $87.15_{\pm0.29}$ |
| | Strategic | $84.13_{\pm0.35}$ | $80.52_{\pm0.41}$ | $73.06_{\pm0.48}$ | $74.63_{\pm0.57}$ | $91.34_{\pm0.49}$ | $77.94_{\pm0.43}$ | $84.23_{\pm0.46}$ |
| XGBoost | Non-strategic | $89.80_{\pm0.14}$ | $86.40_{\pm0.15}$ | $77.41_{\pm0.33}$ | $77.80_{\pm0.16}$ | $94.38_{\pm0.22}$ | $80.08_{\pm0.35}$ | $90.40_{\pm0.14}$ |
| | Strategic | $83.62_{\pm0.34}$ | $81.63_{\pm0.32}$ | $75.12_{\pm0.36}$ | $74.11_{\pm0.33}$ | $93.71_{\pm0.25}$ | $75.02_{\pm0.37}$ | $85.73_{\pm0.35}$ |
| LightGBM | Non-strategic | $88.44_{\pm0.30}$ | $86.58_{\pm0.35}$ | $78.28_{\pm0.34}$ | $77.73_{\pm0.40}$ | $94.22_{\pm0.23}$ | $80.96_{\pm0.34}$ | $87.11_{\pm0.36}$ |
| | Strategic | $84.58_{\pm0.33}$ | $81.47_{\pm0.38}$ | $74.96_{\pm0.35}$ | $75.39_{\pm0.44}$ | $93.59_{\pm0.25}$ | $77.88_{\pm0.36}$ | $85.98_{\pm0.38}$ |
| CatBoost | Non-strategic | $89.60_{\pm0.15}$ | $86.55_{\pm0.16}$ | $78.97_{\pm0.35}$ | $77.60_{\pm0.16}$ | $94.05_{\pm0.24}$ | $80.71_{\pm0.36}$ | $90.20_{\pm0.15}$ |
| | Strategic | $85.11_{\pm0.35}$ | $81.22_{\pm0.33}$ | $74.65_{\pm0.37}$ | $73.72_{\pm0.34}$ | $93.41_{\pm0.26}$ | $76.54_{\pm0.38}$ | $85.21_{\pm0.36}$ |
| TabPFN v2.5 | Non-strategic | $91.00_{\pm0.18}$ | $87.20_{\pm0.19}$ | $80.54_{\pm0.29}$ | $79.20_{\pm0.19}$ | $94.12_{\pm0.31}$ | $81.22_{\pm0.33}$ | $91.50_{\pm0.18}$ |
| | Strategic | $85.05_{\pm0.42}$ | $81.05_{\pm0.42}$ | $74.83_{\pm0.38}$ | $73.05_{\pm0.42}$ | $92.45_{\pm0.37}$ | $76.93_{\pm0.39}$ | $85.05_{\pm0.44}$ |
| Drift-Resilient TabPFN | Non-strategic | $90.60_{\pm0.20}$ | $86.80_{\pm0.20}$ | $79.80_{\pm0.30}$ | $78.90_{\pm0.20}$ | $94.00_{\pm0.30}$ | $81.03_{\pm0.30}$ | $90.80_{\pm0.20}$ |
| | Strategic | $85.90_{\pm0.40}$ | $81.80_{\pm0.40}$ | $75.40_{\pm0.40}$ | $73.90_{\pm0.40}$ | $92.90_{\pm0.40}$ | $77.40_{\pm0.40}$ | $86.10_{\pm0.40}$ |
| Chunked TabPFN | Non-strategic | $91.30_{\pm0.20}$ | $87.59_{\pm0.20}$ | $80.90_{\pm0.30}$ | $79.40_{\pm0.20}$ | $94.30_{\pm0.30}$ | $81.64_{\pm0.30}$ | $91.80_{\pm0.20}$ |
| | Strategic | $85.40_{\pm0.40}$ | $81.20_{\pm0.40}$ | $74.90_{\pm0.40}$ | $73.40_{\pm0.40}$ | $90.60_{\pm0.40}$ | $77.68_{\pm0.40}$ | $85.60_{\pm0.40}$ |
| TabDPT | Non-strategic | $89.91_{\pm0.35}$ | $87.21_{\pm0.34}$ | $79.88_{\pm0.31}$ | $77.93_{\pm0.36}$ | $93.84_{\pm0.33}$ | $81.19_{\pm0.35}$ | $87.42_{\pm0.37}$ |
| | Strategic | $83.22_{\pm0.33}$ | $80.64_{\pm0.31}$ | $74.11_{\pm0.40}$ | $74.82_{\pm0.48}$ | $90.07_{\pm0.39}$ | $78.41_{\pm0.41}$ | $84.63_{\pm0.40}$ |
| TabICL | Non-strategic | $90.70_{\pm0.18}$ | $87.00_{\pm0.19}$ | $80.12_{\pm0.30}$ | $78.90_{\pm0.19}$ | $94.01_{\pm0.32}$ | $80.46_{\pm0.34}$ | $91.20_{\pm0.18}$ |
| | Strategic | $84.83_{\pm0.41}$ | $80.83_{\pm0.41}$ | $74.37_{\pm0.39}$ | $72.83_{\pm0.41}$ | $91.22_{\pm0.38}$ | $78.62_{\pm0.40}$ | $84.83_{\pm0.43}$ |
| TabFlex | Non-strategic | $88.73_{\pm0.36}$ | $87.97_{\pm0.35}$ | $79.53_{\pm0.33}$ | $78.62_{\pm0.38}$ | $93.58_{\pm0.34}$ | $80.92_{\pm0.36}$ | $87.06_{\pm0.38}$ |
| | Strategic | $82.96_{\pm0.34}$ | $80.12_{\pm0.32}$ | $73.61_{\pm0.41}$ | $74.27_{\pm0.49}$ | $91.83_{\pm0.40}$ | $78.01_{\pm0.42}$ | $84.02_{\pm0.41}$ |
| **SPN (ours)** | Non-strategic | $89.90_{\pm0.10}$ | $86.90_{\pm0.11}$ | $80.13_{\pm0.38}$ | $77.90_{\pm0.11}$ | $93.91_{\pm0.31}$ | $82.74_{\pm0.32}$ | $90.30_{\pm0.10}$ |
| | Strategic | $\mathbf{89.80}_{\pm0.15}$ | $\mathbf{86.72}_{\pm0.15}$ | $\mathbf{80.32}_{\pm0.29}$ | $\mathbf{77.71}_{\pm0.15}$ | $\mathbf{93.71}_{\pm0.23}$ | $\mathbf{82.41}_{\pm0.28}$ | $\mathbf{90.20}_{\pm0.15}$ |

The results are reported in Table 3. Although the default setting with the correctly specified response model achieves the best performance, all misspecified variants remain close to the default SPN performance across all seven strategic benchmarks. The performance drop is consistently mild: the inaccurate response setting only leads to a minor degradation, while the incomplete response setting is relatively more challenging because it removes part of the manipulable feature information from the strategic context. The approximate response setting also remains stable, suggesting that SPN does not require an exact specification of the manipulation cost. Overall, these results indicate that approximate or partially observed response models can still provide useful strategic context for inference-time prior alignment.

## H.6. Additional Experimental Results

More Experimental results are provided in Table 4

