# OpenReview forum: "When Tabular Foundation Models Meet Strategic Tabular Data: A Prior Alignment Approach"
_ICML.cc/2026/Conference — ICML 2026 regular_

### Official Review · Reviewer_884v · 2026-02-28

**Soundness:** 3
**Presentation:** 2
**Significance:** 3
**Originality:** 3
**Overall Recommendation:** 3
**Confidence:** 3

**Summary:**

This paper studies a key limitation of existing tabular foundation models: their success largely relies on the assumption that test data are independently and identically distributed (i.i.d.) with respect to the pretraining distribution (i.e., non-strategic settings). However, in real-world applications, once a model is deployed, test samples may adapt their features in response to the decision rule, leading to strategic manipulation and domain shift, which in turn incur prediction bias. To address this issue, the paper proposes a Strategic Prior-data Fitted Network (SPN) that adapts tabular foundation models to strategic environments at inference time. The experimental results demonstrate the effectiveness of SPN across various strategic manipulation scenarios.

**Compliance With Llm Reviewing Policy:**

Affirmed.

**Key Questions For Authors:**

1. To what extent can the proposed method handle strategic manipulation (e.g., under what range or form of manipulation cost)? Does the method have inherent limitations? Is the formulation in Section 3.2 (the equation defining manipulation cost) well justified? The notion of manipulation cost is difficult to understand. In practice, modifying tabular features is often easy for individuals—regardless of which feature is changed or how large the numerical adjustment is.

2. What type of manipulation is applied in Figure 2?

3. How should the label/score y be defined for a manipulated sample? In Line 266, the authors choose to keep the original y. Does this choice make sense, and why? If I understand correctly, when a negative sample is strategically manipulated to achieve a favorable outcome, it may become very similar to a positive sample. In such a case, is it still reasonable to retain the original label?

4. Are the strategic benchmarks used in the experiments based on real-world data? What is the level of distribution mismatch in these settings, for example as measured by uncovered mass?

5. Figure 5 shows the effect of different manipulation proportions. What about the impact of different manipulation strengths on final performance?

6. Why do the authors think SPN is able to maintain performance on standard (non-strategic) tabular benchmarks?

7. What explains the strong performance in Figure 7? Does this suggest that SPN captures the true causal factors in the data?

8. It is strongly recommended to visualize non-strategic and strategic data (e.g., via t-SNE) to provide a more intuitive understanding of the task difficulty. A visualization demonstrating how SPN adapts to distribution shifts would also be helpful.

**Limitations:**

yes

**Strengths And Weaknesses:**

Strengths

1.	The paper claims to be the first to study the generalization of tabular foundation models from non-strategic to strategic domains, which is a highly practical and timely topic.

2.	It characterizes the boundary of PFN-style tabular priors in strategic environments by identifying a fundamental mismatch between non-strategic pretraining and the strategic tabular data encountered at deployment.

3.	The paper provides multiple propositions with corresponding proofs to support the theoretical soundness of the proposed SPN framework. The experimental results also appear strong and generally support the claims.

Weaknesses

1.	The paper is not well organized. The abstract is overly long, while the experimental section is relatively brief. Some subscripts and superscripts in the equations are excessively long and inconsistently formatted (e.g., some include parentheses while others do not), which affects readability.

2.	Some key terms (e.g., manipulation cost) are not clearly defined or sufficiently explained, making parts of the paper difficult to follow. There is also some confusion regarding the specific formulation and implementation of strategic manipulation.

3.	The feasibility of generalizing from non-strategic to strategic settings remains somewhat unclear. For example, in extreme cases where strategic manipulation renders negative samples indistinguishable from positive ones, it is uncertain whether the proposed approach would still be effective.

4.	The experimental evaluation is somewhat limited in scale and diversity, and the paper lacks deeper analysis explaining the reasons behind good or poor performance in different settings.

---

> ### Author Rebuttal · Authors · 2026-03-31
>
> Thank you for your thoughtful comments and suggestions. We would like to address your concerns point by point.
>
> > [W1]: Writing issue.
>
> **Response:**
>
> We thank the reviewer for the helpful feedback on presentation and readability. We make the following revisions (**more details in following comment**):
> - **Abstract.** We have shortened and refined the abstract for improved clarity.
> - **Experiments.** We have expanded the experimental section to improve completeness and presentation.
> - **Notation.** We have simplified and standardized subscripts/superscripts for better readability.
>
> > [W2&Q1]: Clarify the scope and justification of the manipulation cost formulation.
>
> **Response:**
> - **Strategic manipulation.**  In our setting, strategic manipulation refers to individuals adjusting observable features to obtain more favorable decisions. For example, in lending or hiring, individuals may modify controllable attributes based on their understanding of the deployed rule.
> - **Why manipulation cost is meaningful.**
>   - Tabular features typically represent real-world attributes rather than freely editable values. Changing them usually requires effort or resources.
>   - For instance, reducing a debt ratio in a loan application may involve actually repaying debt, which incurs financial cost. The cost function therefore captures the *real-world effort* required for such changes.
>
> > [W3&Q3]: Justify the choice of retaining the original label for manipulated samples.
>
> **Response:**
>
> - **Why keep the original label $y$.**   We follow the standard setting in **strategic classification**, where manipulation corresponds to *gaming the system* rather than improving true qualification.
>   - Even if a manipulated sample $x'$ becomes similar to a positive one, the underlying ground truth does not change.
>   - For example, in loan applications, individuals may temporarily adjust financial indicators to appear qualified, while their true reliability remains unchanged.
>   - Therefore, it is appropriate to keep the original label $y$.
> - **Relation to strategic improvement.** We agree that the reviewer’s example corresponds to **strategic improvement**, where labels may change if manipulation reflects genuine, causally relevant effort (e.g., acquiring skills or education).
>
> > [Q3]: What type of manipulation is applied in Figure 2?
>
> **Response:**
> - The manipulation used in Figure 2 is based on a utility-maximizing strategic response under a Mahalanobis-distance cost.
>   $$
>   b_f(x)=\arg\max_{x'} \; f(x') - \lambda \, c_{\text{Mah}}(x,x').
>   $$
>
> > [Q4] Report distribution mismatch.
>
> **Response:**
> - The datasets used (e.g., Adult, Diabetes) are real-world datasets. The strategic manipulation is simulated, following the strategic learning literature[1,2].
>
> - **Distribution mismatch (uncovered mass).** We quantify the mismatch using an empirical proxy for the uncovered strategic mass. We define
>   $$
>    δ= \frac{1}{N}\sum_{i=1}^N \mathbf{1}[f_{non}(x_i') \neq y_i].
>   $$
>   This measures the fraction of manipulated samples not covered by the non-strategic model.
> - **Empirical results.** We report δ on representative datasets:
>
> **Table 1. Empirical uncovered mass δ.**
>   |Dataset|δ|
>   |-|-|
>   |Adult|0.205|
>   |Census|0.173|
>   |Diabetes|0.214|
>   |Synthetic|0.252|
>   |
>
> > [Q5]: Analyze impact of manipulation strength
>
> **Response:**
> - In Figure 5, the manipulation level is controlled by the **manipulation proportion**, which specifies the fraction of samples subject to strategic manipulation.
> - Therefore, increasing $p$ corresponds to increasing the overall strength of strategic influence at the dataset level.
> - As the proportion of manipulated samples increases, standard methods exhibit clear performance degradation, while SPN remains robust.
>
> > [Q6]: Why SPN maintains performance on tabular benchmarks.
>
> **Response:** SPN does not update model parameters and only performs inference-time adjustment. When the data remain non-strategic, this adjustment is minimal, so the original PFN performance is preserved.
>
> > [Q7] Explains the strong performance in Figure 7.
>
> **Response:** SPN does not capture true causal factors. Instead, the strong performance arises from SPN’s ability to capture **stable response patterns** across related manipulation regimes via in-context learning, leading to improved generalization.
>
> > [Q8]: Visualize non-strategic vs. strategic data.
>
> **Response:**
> Thank you for this insightful suggestion.
> - We visualize both non-strategic and strategic manipulated data using t-SNE on four datasets: Adult, Census-Income (KDD), CDC Diabetes, and PaySim.
> - The results are provided in anonymous repository: https://anonymous.4open.science/r/Visualize-non-strategic-and-strategic-data-371C
>
> > [W4]: Expand experimental evaluation and add deeper analysis.
>
> **Response:**
>
> To expand experimental evaluation and analysis, we have added experiments and discussions above to our revised paper. More experiments are shown in following comments.**

---

> > ### Author Rebuttal · Reviewer_884v · 2026-04-04
> >
> > The rebuttal addresses most questions but remains somewhat shallow. In particular, it lacks technical justification for the manipulation cost and does not clearly address extreme strategic cases (e.g., when manipulated negative samples become indistinguishable from positive ones). It is unclear whether the method can still work in such settings. Therefore, I will keep my score.

---

> > > ### Author Response · Authors · 2026-04-04
> > >
> > > Dear Reviewer 884v,
> > >
> > > We appreciate your detailed questions and thoughtful follow-up. We have prepared thorough and complete responses addressing each of your concerns. **However, due to the 5000-character limit, some technical details had to be compressed.**
> > >
> > > **We would like to provide a more complete clarification focusing on the key issues below.**
> > >
> > > > **Lack technical justification for the manipulation cost.**
> > >
> > > We therefore provide a more complete clarification below.
> > >
> > > **Technical justification of the manipulation cost.**
> > > - The manipulation cost is typically modeled in a Mahalanobis form:
> > >   $$
> > >   c_{Mah}(x,x')=(x'-x)^\top M(x'-x),
> > >   $$
> > >   where $M$ is a positive semidefinite matrix that encodes the covariance structure of the feature space.
> > > - From a modeling perspective, this covariance-aware formulation captures dependencies between features, where changes along different directions incur different costs. It therefore provides a principled way to model strategic effort as **heterogeneous and interdependent**.
> > > - In addition, we would like to kindly note that modeling manipulation cost using quadratic forms (e.g., Mahalanobis distance) is a common and well-established practice in the strategic learning literature[1,2,3], as it provides a tractable way to capture heterogeneous and correlated feature changes.
> > >
> > > **Empirical validation of the manipulation cost choice:** To further evaluate whether this choice is appropriate, we supplement two experiments.
> > > - **Manipulation cost mismatch.** In our default SPN setup, the strategic context is constructed using a Mahalanobis-based manipulation;
> > >   - We consider three variants to evaluate the robustness:
> > >     - **Inaccurate:** add noise to the default cost;
> > >     - **Incomplete:** only a subset of features is manipulated;
> > >     - **Approximate:** replace the Mahalanobis cost with a Euclidean cost;
> > >   - We report the corresponding accuracy below:
> > >
> > > **Table 1. Performance under mismatch cost**
> > > |Settings|Adult|German|Diabetes|
> > > |-|-|-|-|
> > > |Default|83.1|88.0|82.2|
> > > |Ina.|82.7|87.6|81.8|
> > > |Inc.|81.9|86.9|81.0|
> > > |Appro.|82.2|87.2|81.4|
> > > |
> > >
> > > - **Evaluation on TableShift.**
> > >   - We also supplement experiments on the TableShift benchmark[4], **where feature changes are governed by a known manipulation process or cost**.
> > >   - We report ID accuracy(%), OOD accuracy(%), and Shift gap(%) on College Scorecard task.
> > >
> > > **Table 2. Performance on TableShift.**
> > > |Model|ID Acc|OOD Acc|Shift Gap|
> > > |-|-|-|-|
> > > |FT-Transformer|93.4|84.4|9.0|
> > > |MLP|91.8|82.9|8.9|
> > > |TabPFN|93.2|83.8|9.4|
> > > |SPN(ours)|93.1|86.6|**6.5**|
> > > |
> > >
> > > - **Analysis.** These results support the reasonableness of the proposed manipulation cost. SPN remains effective under manipulation mismatch and continues to perform well under uncertain shifts (TableShift), indicating that the Mahalanobis cost serves as a **structured approximation** of realistic feature manipulation.
> > >
> > > > **Analysis of the extreme strategic cases.**
> > >
> > > We agree that in extreme cases, manipulated negative samples may become indistinguishable from positive ones. To better understand how our method behaves in this regime, we conduct a simulation study.
> > >
> > > - **Formalization of the extreme regime.**
> > >   To explicitly study this scenario, we construct a controllable manipulation process:
> > >   $$
> > >   x'=(1-α)x+αx^+,
> > >   $$
> > >   where $x$ is a negative sample, $x^+$ is a randomly sampled positive instance, and $α\in [0,1]$. As $α\to 1$, the manipulated negative distribution converges to the positive distribution, leading to statistical indistinguishability.
> > > - **Experimental evaluation.**
> > >   - We evaluate SPN under increasing manipulation strength $α\in \\{0.5,0.8,0.9,1.0\\}$, progressively approaching the indistinguishable regime.
> > >   - We report AUC under both a **balanced setting** (positive samples: negative samples=1:1) and an **unbalanced setting** (1:3).
> > >
> > > **Table 3. Results on Balanced Setting.**
> > > |α|TabPFN|SPN(ours)|
> > > |-|-|-|
> > > |0.5|78.9|83.3|
> > > |0.8| 65.2|76.8|
> > > |0.9|53.7|68.5|
> > > |1.0|50.6|53.9|
> > > |
> > >
> > > **Table 4. Results on Unbalanced Setting.**
> > > |α|TabPFN|SPN(ours)|
> > > |-|-|-|
> > > |0.5|67.9|81.1|
> > > |0.8|63.4|74.2|
> > > |0.9|52.9|66.0|
> > > |1.0|49.2|53.1|
> > > |
> > >
> > > - **Analysis.**
> > >   - Under both balanced and unbalanced settings, AUC decreases as α increases, indicating that the task becomes substantially harder as manipulated negatives approach the positive class.
> > >   - SPN consistently outperforms the baseline at intermediate levels of α and shows a more gradual degradation. When α=1.0, both methods approach chance level, reflecting the difficulty of the extreme case.
> > >
> > >
> > > We thank the reviewer for the helpful comments, which have improved the quality of our work. All revisions mentioned above have been incorporated into our revised paper.
> > >
> > > ---
> > > *Refs:*
> > >
> > > [1] Hardt et al. Strategic classification.
> > >
> > > [2] Chen et al. Learning to Incentivize Improvements from Strategic Agents
> > >
> > > [3] Ghalme et al. Strategic Classification in the Dark
> > >
> > > [4] Gardner et al. Benchmarking Distribution Shift in Tabular Data with TableShift

---

### Official Review · Reviewer_ejjy · 2026-03-04

**Soundness:** 2
**Presentation:** 3
**Significance:** 2
**Originality:** 2
**Overall Recommendation:** 4
**Confidence:** 3

**Summary:**

The paper studies PFN-style tabular foundation models under strategic settings where agents manipulate features after deployment, inducing distribution shift between training and inference. It identifies a meta-prior mismatch between non-strategic PFN pretraining and strategic task distributions, which leads to prediction bias, and proposes Strategic Prior-data Fitted Networks (SPN), an inference-time approach that constructs strategic contexts and leverages in-context learning to approximate post-manipulation inputs. Experiments on several tabular datasets show improved robustness under manipulation while maintaining performance in non-strategic settings. However, the proposed SPN largely appears to be a context augmentation strategy rather than a fundamentally new learning method.

**Compliance With Llm Reviewing Policy:**

Affirmed.

**Final Justification:**

The final score is updated.
Thanks!

**Key Questions For Authors:**

1- The paper studies robustness to distribution shift but does not evaluate on standard tabular shift benchmarks such as **TableShift**. Can the authors evaluate SPN on these datasets? Does the proposed context construction mechanism still improve robustness under non-strategic covariate shifts? If the method relies on simulated manipulation functions $b_f(x)$, how would SPN operate in TableShift settings where the shift mechanism is unknown?

2- The theory introduces the uncovered strategic mass $\delta$. Can the authors provide empirical estimates of $\delta$ for the datasets used in the experiments? Without such estimates, it is unclear how informative the theoretical bias bound is in practice.

3- All experiments focus on binary classification. Does the SPN framework extend naturally to regression tasks, which are also common in strategic tabular settings?

4- SPN requires access to $b_f(\cdot)$ to construct the strategic context. How would $b_f$ be estimated or approximated in practice when it is unknown? Have the authors evaluated SPN under *misspecified manipulation models*, where the context uses an approximation of $b_f$ rather than the true function?

**Limitations:**

A key limitation is that SPN assumes access to the manipulation function $b_f(\cdot)$ to construct the strategic context, which may be unavailable or difficult to estimate in practice. In addition, the theoretical guarantees are asymptotic and do not provide actionable finite-sample bounds.

**Strengths And Weaknesses:**

***Strengths***

• The paper targets the relatively unexplored intersection of tabular foundation models (e.g., PFNs, TabPFN-v2.5, TabICL, TabDPT, TabFlex) and strategic tabular decision-making, which is practically relevant for applications such as credit scoring, spam detection, and policy allocation.

• The work provides a formal characterization of meta-prior mismatch (Section 4), derives a lower bound on unavoidable prediction bias (Proposition 4.4), and shows that SPN reduces the uncovered strategic mass (Proposition 5.3). The theoretical arguments appear complete and follow standard minimax/Le Cam-style reasoning.

***Weaknesses***

• The experimental comparisons mainly involve non-strategic tabular models and PFN variants, along with a fine-tuning vs. ICL cost case study, but do not include strong strategy-aware baselines (e.g., recent strategic classification methods designed to be robust under manipulation).

• Most strategic experiments report accuracy only, despite strategic settings being particularly sensitive to false positives/negatives and fairness concerns. Apart from one figure reporting false positive error, there is limited analysis of calibration, welfare, or fairness under manipulation.

• Although SPN consistently outperforms baselines under strategic manipulation, the paper does not analyze why SPN accuracy itself still degrades (albeit less) as the manipulation proportion increases (Figure 5). Understanding these residual failure modes would strengthen the work.

• There appear to be inconsistencies between the paper and the released code/supplementary materials (e.g., manipulation generation and experimental setup), which makes it unclear whether the reported results can be fully reproduced.

---

> ### Author Rebuttal · Authors · 2026-03-31
>
> Dear Reviewer ejjy,
>
> Thank you for your thoughtful comments and suggestions. We would like to address your concerns point by point.
> > [W1]: Supplement strategic-aware baselines.
>
> **Response:**
> - We have added comparisons with strategic-aware methods[1,2,3,4].
> - The results are summarized below (**see more details and results in our following comments dur to limited space**).
>
> **Table 1. Performance  (accuracy %).**
> |Method|Adult|Spambase|German|
> |-|-|-|-|
> |[1]|0.816|0.737|0.881|
> |[2]|0.811|0.733|0.874|
> |[3]|0.829|0.741|0.888|
> |[4]|0.831|0.743|0.892|
> |SPN(ours)|0.834|0.761|0.897|
> |
> - SPN with ICL achieves consistently performance with strategy-aware training methods.
>
> > [W2]: Add analysis of false positives/negatives, fairness, and welfare metrics.
>
> **Response:**
> - We have extended our experiments to include additional metrics covering error decomposition (FPR/FNR), fairness (measured via demographic parity), and welfare under strategic manipulation (**more details and results in following comments**).
>
> **Table 2. Performance on German dataset.**
> |Method|FPR↓|FNR↓|DP Gap↓|Welfare↑|
> |-|-|-|-|-|
> |[4]|0.171|0.236|0.082|0.637|
> |TabPFN|0.182|0.247|0.094|0.621|
> |SPN(ours)|0.149|0.221|0.061|0.658|
> |
> - These results suggest that SPN achieves consistent improvements across accuracy, error rates, and fairness-related metrics.
>
> > [W3]: Analyze why SPN accuracy degrades as manipulation proportion increases.
>
> **Response:**
> - In Figure 5, the performance drop mainly occurs at intermediate manipulation proportions (around 0.5), where the test distribution becomes a mixture of original and manipulated samples.
> - These two types of samples follow different feature-label relationships, making the induced distribution less coherent and introducing ambiguity in the strategic patterns inferred by SPN.
>
> > [W4] Inconsistencies between settings and code.
>
> **Response:** We have revised the manuscript to clarify the experimental setup and would like to release the full code to ensure reproducibility.
>
> > [Q1]: Evaluate SPN on TableShift benchmarks.
>
> **Response:**
> - We have added experiments on representative TableShift tasks, following the standard protocol and reporting ID accuracy, OOD accuracy, and shift gap (**more details and results in following comment**).
>
> **Table 3. Performance on College Scorecard task.**
> |Method|ID Acc|OOD Acc|Shift Gap↓|
> |-|-|-|-|
> |CatBoost|94.0|85.2|8.8|
> |FT-Transformer|93.4|84.4|9.0|
> |TabPFN|93.2|83.8 |9.4|
> |SPN (ours)|93.1|86.6|6.5|
> |
> - In TableShift, SPN does not rely on a true manipulation function; instead, the constructed context serves as a perturbation-based augmentation for adapting to distribution shifts.
>
> > [Q2]: Provide empirical estimates of the uncovered strategic mass δ.
>
> **Response:**
>
> To make the uncovered strategic mass δ empirically interpretable, we estimate it using a sample-level proxy (more details in following comment):
> $$
> δ = \frac{1}{N}\sum_{i=1}^N\mathbf{1}[f_{non}(x_i')\neq y_i].
> $$
> **Table 4. δ under different manipulation proportions.**
> |Dataset|δ@0.5|δ@0.8|δ@1.0|
> |-|-|-|-|
> |Adult|0.162|0.191|0.205|
> |Census(KDD)|0.118|0.152|0.173|
> |Diabetes|0.178|0.201|0.214|
> |Synthetic|0.102|0.134|0.152|
> |
>
> > [Q3]: Extend SPN framework to regression tasks.
>
> **Response:**
> - We supplement the regression experiments (**more details and results in following comment**).
>   - Metric: MSE.
>   - Datasets: California Housing and Bike Sharing.
>
> **Table 5. Non-strategic Performance**
> |Model|California|Bike|
> |-|-|-|
> |CatBoost|0.265|0.116|
> |TabPFN|0.268|0.112|
> |SPN(ours)|0.269|0.115|
> |
>
> **Table 6. Strategic Performance**
> |Model|California|Bike|
> |-|-|-|
> |[4]|0.283|0.126|
> |TabPFN|0.311|0.156|
> |SPN(ours)|0.273|0.129|
> |
> - SPN remains lower MSE, indicating that the framework generalizes effectively to regression tasks.
>
> > [Q4]: Evaluate SPN under misspecified manipulation models
>
> **Response:**
> - In practice, $b_f$ can be approximated from observed pre- and post-adjustment behavior, but is difficult to recover precisely.
> - To study this, we evaluate SPN under misspecified manipulation models.
> - We use a Mahalanobis-based manipulation as the default, fix it at test time, and construct the ICL context using misspecified models (inaccurate, incomplete, and approximate), **more details in following comments**.
>
> **Table 7. Performance (accuracy, %).**
> |Case|Adult|Census|Diabetes|
> |-|-|-|-|
> |Default|83.1|88.0|82.2|
> |Inaccurate|82.7|87.6|81.8|
> |Incomplete|81.9|86.9|81.0|
> |Approximate|82.2 |87.2|81.4|
> |
> - SPN degrades only moderately under misspecified manipulation models.
>
> **We have added these experiments and discussions to our revised paper.**
>
> > Limitation:  provide actionable finite-sample bounds.
>
> **Response:**  Due to space limitations, we provide corresponding finite-sample bounds in the following comments.
>
> ----
> [1] Strategic Classification
>
> [2] The Social Cost of Strategic Classification
>
> [3] Strategic Classification Made Practical
>
> [4] Who Leads and Who Follows in Strategic Classification

---

> > ### Author Rebuttal · Reviewer_ejjy · 2026-04-02
> >
> > Thank you for the clarification. I am leaning toward increasing my score from 3 to 4. It would be helpful if you could provide additional experimental results or analysis, as discussed in the rebuttal, to further support your claims..

---

> > > ### Author Response · Authors · 2026-04-02
> > >
> > > Dear Reviewer,
> > >
> > > We appreciate your feedback. **We provide additional clarifications and experiments below to help verify that our revisions fully address your concerns.**
> > > > **Weakness 1**
> > >
> > > **Supplement:**
> > > - Experimental settings:
> > >     - For strategy-aware baselines, we follow the standard supervised protocol: each method is trained on 80% of the data and evaluated on the remaining 20%.
> > >     - SPN operates in an ICL manner, where 10% of the data is used to construct the strategic context, and evaluation is performed on the same test split.
> > > - Additional experimental results：
> > >     - We evaluate on two additional datasets: Diabetes and Synthetic.
> > >
> > > **Table 1. Comparison with representative strategy-aware baselines (accuracy).**
> > > |Method|Diabetes|Synthetic|
> > > |-|-|-|
> > > |[1]|0.803|0.889|
> > > |[2]|0.812|0.886|
> > > |[3]|0.810|0.890|
> > > |[4]|0.816|0.897|
> > > |SPN|0.820|0.901|
> > >
> > > - **Analysis.** Across all datasets, SPN achieves performance comparable to these representative strategy-aware baselines. Notably, this is achieved in an inference-time ICL setting without any parameter updates.
> > >
> > > > **Weakness 2**
> > >
> > > **Supplement:**
> > > - Experimental settings: We follow the same strategic setting as in the main experiments, with strategic manipulation generated via a Mahalanobis-based response model.
> > >     - FPR, FNR, and welfare are evaluated on post-manipulation predictions.
> > >     - Fairness is measured using demographic parity (DP gap) between groups.
> > > - Additional experimental results:
> > >     - We evaluate on two additional datasets: Adult and Diabetes.
> > >
> > > **Table 2. Performance on *Adult*.**
> > > |Method|FPR↓|FNR↓|DP Gap↓|Welfare↑|
> > > |-|-|-|-|-|
> > > |TabPFN|0.214|0.279|0.093|0.571|
> > > |Hardt et al. (2016)[1]|0.182|0.251|0.102|0.583|
> > > |Levanon et al. (2021)[3] |0.176|0.243|0.096|0.598|
> > > |SPN(ours)|0.176|0.241|0.095|0.596|
> > >
> > > **Table 3. Performance on *Diabetes*.**
> > > |Method|FPR↓|FNR↓|DP Gap↓|Welfare↑|
> > > |-|-|-|-|-|
> > > |TabPFN|0.142|0.174|0.063|0.726|
> > > |Hardt et al. (2016)[1]|0.131|0.162|0.071|0.742|
> > > |Levanon et al. (2021)[3]|0.120|0.153|0.063|0.759|
> > > |SPN|0.118|0.149|0.065|0.766|
> > >
> > > - **Analysis.** SPN achieves comparable performance to strategy-aware baselines across all metrics, while operating without retraining.
> > >
> > > > **Question 1**
> > >
> > > **Supplement:**
> > > - In addition to the results reported above, we also evaluate SPN on the Income task.
> > >
> > > **Table 4. Performance on *Income task* from TableShift.**
> > > |Method|ID Acc(%)| OOD Acc(%)|Shift Gap↓|
> > > |-|-|-|-|
> > > |CatBoost|82.2|80.8|1.4|
> > > |FT-Transformer|81.8|80.1|1.7|
> > > |MLP|81.0|79.1|1.9|
> > > |TabPFN|81.9| 80.2|1.7|
> > > |SPN|82.0|80.9|**1.1**|
> > >
> > > - **Analysis.** SPN achieves comparable performance on TableShift benchmarks and remains effective under covariate shift without requiring an explicit manipulation function.
> > >
> > > > **Question 2**
> > >
> > > **Supplement:** The Details of uncovered strategic mass estimation.
> > > - The uncovered strategic mass $\delta$ is defined at the meta-prior level and is not directly observable from finite samples.
> > > - To make it empirically interpretable, we approximate it using the failure rate of a non-strategic model on strategically manipulated inputs.
> > > - Specifically, let $x_i'=b_f(x_i)$ denote the manipulated sample and $f_{non}$ the non-strategic model. We define
> > > $$
> > > δ=\frac{1}{N}\sum_{i=1}^N \mathbf{1}[f_{non}(x_i')\neq y_i].
> > > $$
> > > - This quantity serves as a practical proxy for the mismatch induced by strategic manipulation.
> > >
> > > > **Limitation**: provide actionable finite-sample bounds.
> > >
> > > **Response:**
> > > - The uncovered strategic mass can be estimated in finite samples via the empirical proxy $δ$ defined above.
> > > - Since $δ$ is an average of Bernoulli indicators over manipulated samples, standard concentration results (e.g., Hoeffding’s inequality) imply that, for any $\epsilon>0$,
> > > $$
> > > \Pr(|δ-\mathbb{E}[δ]|\ge\epsilon)\le 2\exp(-2N\epsilon^2).
> > > $$
> > > - Thus, $δ$ concentrates around its population counterpart at rate $O(\sqrt{1/N})$, ensuring that the central quantity driving the bias can be reliably estimated in finite-sample regimes. This makes the theoretical characterization informative in practice.
> > >
> > > > **Question 3**
> > >
> > > **Supplement:**
> > > - Additional experimental results with *Housing* and *Energy* dataset in the strategic setting:
> > >
> > > **Table 5. Performance on regression tasks (MSE↓).**
> > > |Method|Housing|Energy|
> > > |-|-|-|
> > > |TabPFN|0.612|0.428|
> > > |[1]|0.541|0.361|
> > > |[4]|0.536|0.357|
> > > |SPN(ours)|0.533|0.355|
> > >
> > > > **Question 4**
> > >
> > > **Supplement:** Details of misspecified manipulation.
> > > - Default: Mahalanobis-based manipulation function.
> > > - Inaccurate: add noise to default function.
> > > - Incomplete: only a subset of features is manipulated.
> > > - Approximate: replace Mahalanobis cost with Euclidean cost.
> > >
> > >
> > > **We thank the reviewer for the helpful comments, which have improved the quality of our work. All revisions have been incorporated into revised paper.**
> > >
> > > ---
> > > [1] Hardt et al. Strategic Classification
> > >
> > > [2] Milli et al. The Social Cost of Strategic Classification
> > >
> > > [3] Levanon et al. Strategic Classification Made Practical
> > >
> > > [4] Zrnic et al. Who Leads and Who Follows in Strategic Classification?

---

### Official Review · Reviewer_Sg8N · 2026-03-12

**Soundness:** 3
**Presentation:** 4
**Significance:** 3
**Originality:** 3
**Overall Recommendation:** 5
**Confidence:** 4

**Summary:**

This paper studies tabular foundation models, specifically in strategic tabular settings where deployed classifiers face feature manipulation and therefore post-deployment distribution shift. The paper argues that PFNs are pretrained under a non-strategic meta-prior, while deployment in strategic environments induces task distributions outside that prior’s support, creating systematic prediction bias. To address this, the authors propose Strategic Prior-data Fitted Networks (SPN), an inference-time method that constructs a strategic in-context table by pairing original examples with simulated post-manipulation counterparts, then runs the PFN on this modified context without retraining or architecture changes. Empirically, the paper reports that vanilla PFNs degrade as manipulation increases, while SPN is more stable under several manipulation regimes and largely preserves non-strategic accuracy on standard tabular benchmarks.

**Compliance With Llm Reviewing Policy:**

Affirmed.

**Final Justification:**

The paper targets an underexplored problem and represents the first attempt to improve foundation models for strategic classification. I believe it has the potential to inspire future work and recommend acceptance.

**Key Questions For Authors:**

- How sensitive is SPN to misspecification of the manipulation model?
- Can the authors better justify the connection between the abstract manipulation regimes and realistic strategic behavior on the benchmark datasets?
- Can you elaborate on limitations of the proposed method?

**Limitations:**

yes

**Strengths And Weaknesses:**

### Strengths
- Tackles an important and underexplored problem and provides a clear high-level motivation.
- Very well-organized content and an easy-to-follow structure.
- The proposed SPN method is practically appealing because it adapts the model at inference time, without architectural changes, or new training paradigms.
- Empirical results appear strong. The authors convincingly show that standard models degrade as manipulation increases, while SPN remains more stable in high-manipulation settings.
- The paper also shows that SPN does not substantially hurt non-strategic accuracy, which is important for practical adoption.
- The evaluation covers several useful axes, including different manipulation regimes, varying proportions of manipulated inputs, and the effect of the number of in-context examples.
- Evaluation includes models beyond PFNs.
- The authors’ decisions are well motivated by ablations, such as the case study in Section 5.1, which supports the decision to focus on ICL over fine-tuning.
- To my knowledge, this is the first attempt to improve foundation models for strategic classification, and I believe it has the potential to inspire future work.

### Weaknesses
- The discussion of limitations is insufficient. The societal impact discussion also feels too light for a paper about strategic behavior in domains such as credit or public decision-making, where modeling errors could have important fairness and deployment consequences.
- The paper does not clearly connect the studied manipulation regimes to plausible real-world counterparts. For example, it would help to explain what kinds of strategic behavior in practice would correspond to a Mahalanobis-style manipulation model.
- What also seems missing is a robustness analysis to misspecification of the manipulation model \(b_f\). SPN explicitly constructs strategic in-context examples using simulated post-manipulation points \(x_i' = b_f(x_i)\), and evaluation is also defined through such response maps, but the paper mainly studies performance under several *assumed* manipulation regimes rather than testing what happens when the available \(b_f\) is inaccurate, incomplete, or only approximately known. This is important because the practical usefulness of SPN may depend heavily on how well the manipulation process is modeled.

---

> ### Author Rebuttal · Authors · 2026-03-31
>
> Dear Reviewer Sg8N,
>
> Thank you for your thoughtful comments and suggestions. We would like to address your concerns point by point.
>
> > [W3&Q1]: How sensitive is SPN to misspecification of the manipulation model?
>
> **Response:**
>
> We thank the reviewer for this important question.
> - In our default SPN setup, the strategic context is constructed using a Mahalanobis-based best-response function:
> $$
> b_f(x)=\arg\max_{x'} f(x') - \lambda c_{\text{Mah}}(x,x').
> $$
>
> - To evaluate robustness under misspecified manipulation models, we keep the **test-time strategic behavior** fixed as the standard Mahalanobis-based response $b_f$, and instead construct the ICL strategic context using misspecified variants $\tilde{b}_f$. We consider the following three cases:
>   - **Inaccurate response.**
>     We first compute the standard best response $b_f(x)$, and then add perturbation to model imperfect understanding or execution of the manipulation:
>     $$
>     \tilde{b}_f^{inaccurate}(x) = b_f(x) + \epsilon,  \quad \epsilon \sim \mathcal{N}(0,\sigma^2 I).
>     $$
>   - **Incomplete response.**
>     We assume that only a subset of manipulable features is captured by the available response model:
>     $$
>     \tilde{b}_f^{incomplete}(x)=  m \odot b_f(x) + (1-m) \odot x,
>     $$
>     where m is a random binary mask and $\odot$ denotes element-wise multiplication.
>   - **Approximately known response.**
>     We replace the true Mahalanobis cost with a simplified Euclidean cost and define:
>     $$
>     \tilde{b}_f^{approx}(x)= \arg\max _{x'} f(x') - \lambda  \|x'-x\|_2^2,
>     $$
>     where the model captures the general structure but does not match the true manipulation process.
>
> - We report the accuracy under these settings with three different datasets:
>
>
> |Settings|Adult|German|Diabetes|
> |-|-|-|-|
> |Default|83.1|88.0|82.2|
> |Inaccurate|82.7|87.6|81.8|
> |Incomplete|81.9 |86.9|81.0|
> |Approximate|82.2|87.2|81.4|
> |
>
> - **Analysis.** We observe that while the accurate setting achieves the best performance, all misspecified variants remain close, with only minor degradation. This indicates that SPN is not overly sensitive to the exact specification of the manipulation model.
>
> We have added these experiments to our revised paper.
>
> > [W2&Q2] Can the authors better justify the connection between the abstract manipulation regimes and realistic strategic behavior on the benchmark datasets?
>
> **Response:**
>
> We agree that the connection between the abstract manipulation regimes and realistic strategic behavior should be stated more clearly.
>
> - In real-world strategic settings, individuals adjust their attributes by trading off the potential benefit of a favorable decision against the effort required for such changes.
> - Our manipulation regimes capture this trade-off through the choice of a cost function, which abstracts how difficult different attributes are to modify.
> - In practice, feature adjustments are often both heterogeneous and correlated.
>   - For example, a Euclidean cost assumes relatively uniform and independent adjustments, while a Mahalanobis-style cost captures feature-dependent difficulty and correlations through the covariance structure.
> - As a concrete example:
>   - In the Adult dataset, features such as education, occupation, and working hours can be viewed as income-related attributes.
>   - Strategic behavior in this setting may correspond to individuals attempting to improve these attributes to increase their chance of being classified as high income.
>   - Such adjustments are not equally easy, nor are they fully independent, which makes a structured cost model a reasonable abstraction.
>
> We have revised the paper to make this connection more explicit.
>
> > [W1&Q3]: The discussion of limitations is insufficient.
>
> **Response:**
> - We agree that, in strategic decision-making settings such as credit or public allocation, modeling errors may have **important implications for fairness and resource allocation**. Since individuals may differ in their access to information or ability to adapt, strategic responses can interact with decision rules and lead to disparities across groups.
> - We acknowledge these risks in realistic deployments. In such settings, several directions may help mitigate potential issues:
>   - **Incorporating robustness considerations into the modeling process**, for example, by evaluating model behavior under varying strategic assumptions or by adopting uncertainty-aware approaches to reduce sensitivity to misspecification.
>   - **Accounting for heterogeneity in individuals’ ability to respond strategically**, which may help reduce unintended disparities and improve fairness in downstream decisions.
>   - More broadly, these considerations highlight the importance of carefully evaluating strategic models under diverse assumptions before deployment.
>
> We have revised the paper to include a clearer discussion of these limitations and their deployment implications. A more detailed version is included in the following comments due to limited space.

---

> > ### Author Rebuttal · Reviewer_Sg8N · 2026-04-02
> >
> > The rebuttal addresses most of my concerns. I therefore maintain my positive score.

---

> > > ### Author Response · Authors · 2026-04-02
> > >
> > > Dear Reviewer Sg8N,
> > >
> > > We sincerely thank you for your encouraging feedback.
> > >
> > > In the revised version of the paper, we will carefully incorporate your suggestions to further improve the presentation of our work.

---

### Decision · Program_Chairs · 2026-04-30

**Decision:**

Accept (regular)

**Comment:**

This paper addresses how feature manipulation breaks tabular foundation models (PFNs). The authors introduce Strategic PFNs (SPN), an inference-time fix that adds simulated manipulated data to the model's context. SPN stabilizes performance against adversarial manipulation without requiring any model retraining.

Interesting and timely research direction. This is a typical example of a paper whose content improved during the discussion period. The authors are compelled to incorporate the extra evidence they provided during this period into the camera-ready version of this paper.

Note: Leaving the initial score unchanged was an oversight on the part of Reviewer ejjy.